**Data Availability Statement:** All relevant data are within the manuscript and its Supporting information files.

# Experimental and computer study of the mechanism and identification of conditions for energy-efficient removal of moisture from materials under ultrasonic exposure

Vladimir Khmelev, Andrey Shalunov�*, Sergey Terentiev�, Roman Golykh, Viktor Nesterov�

Biysk Technological Institute (Branch) Altai State Technical University, Biysk, Russia

* shalunov@u-sonic.ru

## Abstract

The article is devoted to investigation of energy-efficient moisture removal from capillary-porous materials. Moisture is removed by dispersion at collapse of cylindrical cavitation bubbles, formed by ultrasonic vibrations in the capillaries of the material. Mathematical model, which allowed to investigate the mechanism of moisture dispersion, has been developed. Necessity of realization of cavitation bubble full life cycle in capillary (slow growth, rapid expansion with deformation, collapse) was found. An optimal range of sound pressure levels from 150 dB ("critical level" at which dispersion of water from capillary starts) up to 170 dB (dispersion productivity growth stops due to cavitation bubbles reaching maximum size equal to diameter of capillary) was determined. It is shown that the size of the dewatered sample for maximum drying efficiency should correspond to the ultrasonic wavelength in air. Ultrasonic dispersion of liquid during drying was confirmed experimentally. It is found that for significant reduction of drying time (up to 50% and more) it is necessary to affect in the range of 165–170 dB. And the materials to be dried must be placed as particles or layers having dimensions or thicknesses corresponding to the length of the ultrasonic wave in air. The implementation of ultrasonic drying, on the example of food products (beets) provided a reduction in drying time of 1.9 times, while reducing energy costs by 1.7 times in comparison with convective drying.

## Introduction

Drying is required to ensure the long-term storage of products with the following rehydration for application. Typically, this is a long and energy-intensive process. One of the ways to increase the efficiency (increase the speed and reduce energy consumption) of drying is to expose the material being dried to acoustic vibrations in the audible or ultrasonic (US) frequency range. Such vibrations are capable of introducing into materials energies proportional to the square of the frequency of the vibrations used.

**Funding:** The study was supported by a grant from Russian Science Foundation No. 21-79-10359, https://rscf.ru/en/project/21-79-10359/.

**Competing interests:** The authors have declared that no competing interests exist.

The results of the drying process acceleration due to the exposure to acoustic vibrations was first experimentally determined and confirmed in the middle of the twentieth century [1–3]. In 1955, P. Greguss [4] demonstrated the drying acceleration of wet cotton fiber under the contact exposure to ultrasonic vibrations with a frequency of 25 kHz. Since then, research into the effectiveness of drying various materials has been carried out continuously. The reason for this is that during ultrasonic drying there is no significant increase in the temperature of the dried product. Thanks to this, drying objects do not lose their properties, and after drying, they are restored in structure and shape almost to their original state. Among the latest achievements in the field of ultrasonic drying, it is worth noting the work of, for example, Gallego-Juarez and Sun in which a reduction in drying time for carrots by 31% at a sound pressure level of 155 dB and a temperature of 60°C [5] and up to 32% when drying the sewage sludge at an ultrasonic vibration power of 90 W and a temperature of 70°C [6].

Consolidation of the available research results devoted to drying of the materials with various structure and properties in various modes and exposure conditions indicates that the drying acceleration begins from a certain sound pressure level, the value of which, according to the available data, exceeds 125–150 dB [7, 8]. In this regard, many authors use the concept of "critical sound pressure level" that refers to the sound pressure level, below which the ultrasonic exposure has almost no impact on the drying process [7, 9].

Unfortunately, until the present time, there has been no unbiased information about the ultrasonic impact parameter value, at which a significant (step-like) drying process intensification is commenced. The lack of such data is due to the insufficient level of knowledge about the drying efficiency improvement mechanisms when the sound pressure level exceeds a certain "critical" value.

To explain the experimentally observed acceleration of the drying process due to the impact of ultrasonic vibrations on the dried material, various previously proposed mechanisms for accelerated moisture evacuation were subject to critical analysis. Among them, the acoustic flows near the material surface or the "vibration effect" are considered the most effective ones. This effect was used to explain the main vibration impact on acceleration of the drying process [10].

In addition, the thermal effect was considered as an intensifying mechanism, since the ultrasonic vibration damping causes slight heating in the material. However, many authors [10–13] do not assign a serious role to this effect. According to some authors, a significant influence is exerted by the repeated change sin pressure near the material surface that leads to the so-called "sponge effect" [14–16]. However, the existing mechanisms cannot explain the availability of clear boundaries in terms of the exposure intensity at the level of which an abrupt acceleration of the drying process is shown.

This can only be explained by the assumption (made by the authors of papers [17, 18]) that during the ultrasonic drying process, moisture is evacuated from the surface of material being dried by its mechanical atomization (dispersion) under the impact of vibrations. However, the moisture dispersion (atomization) mechanism from a solid during the drying process has not been studied at the theoretical or practical level. In [19], it has been assumed that dispersion is a consequence of the collapse of cavitation bubbles ("cavitation mechanism"). This assumption is based on the fact that the ultrasonic vibrations can cause cavitation in a fluid medium inside a porous material [20, 21]. In this case, the possible collapse of cavitation bubbles can lead to the significant local increases in pressure and temperature [22] and ensures the liquid evacuation from the dried material in the form of small drops, i.e. without its conversion to the steam. Obviously, the implementation of such a process can significantly reduce both the process duration and energy consumption for drying materials.

However, the cavitation acceleration mechanism of the drying process has not yet been studied and confirmed. This does not allow to theoretically determine the boundaries of

occurrence of the ultrasonic moisture dispersion (moisture evacuation without a phase transformation: water to steam conversion due to heating), i.e. determine the critical sound pressure level at which drying efficiency increases significantly. It is also obvious that an increase in the sound pressure level of ultrasonic vibrations requires an increase in the cost of such vibrations. Therefore, there is a need to determine the maximum level of ultrasonic exposure, at which the ultrasonic drying process remains not only efficient, but also economically sound.

Thus, the need to increase the efficiency of drying through theoretical and experimental confirmation of the mechanism of cavitation dispersion of moisture, and determining the range of sound pressure levels at which this mechanism is realized, determines the relevance of research.

## Theoretical studies of ultrasonic material desiccation

Identification and interpretation of the moisture evacuation mechanisms without the phase transformation of water into steam under the influence of ultrasonic vibrations is impossible without due regard to the structure and properties of materials being dried, the dimensions of samples being dried, or the dimensions of layers being dried.

### Stages of energy conversion from the ultrasonic vibrations into the material desiccation

When developing a physical and mathematical model of the possible liquid dispersion process from the pores and capillaries, all stages of the energy conversion of ultrasonic vibrations were considered, namely:

1. energy conversion of ultrasonic gas vibrations into the solid phase vibrations;

2. energy conversion of the solid phase vibrations into the liquid vibrations located in the capillary cavity;

3. energy conversion of the liquid vibrations into the expansion (including the stages of slow and rapid growth) of bubbles located in the capillaries with their deformation;

4. conversion of energy stored in the bubbles into the energy of shock waves propagating along the capillaries;

5. energy conversion of the shock waves into the kinetic energy of liquid inside the capillary;

6. conversion of the liquid kinetic energy into an increase in the free surface.

The stages of energy conversion are shown in simplified form in Fig 1.

Further, each stage of energy conversion of ultrasonic vibrations has the relevant developed submodels described in the following sections.

### Submodel of ultrasonic wave energy conversion into the vibrations of solid and liquid phases

The submodel of primary ultrasonic wave energy conversion into the vibrations of the solid and liquid phases is designed to identify the parameters of acoustic field generated in the solid and liquid phases at the given parameters of primary ultrasonic field generated in a gaseous medium near the dried sample surface.

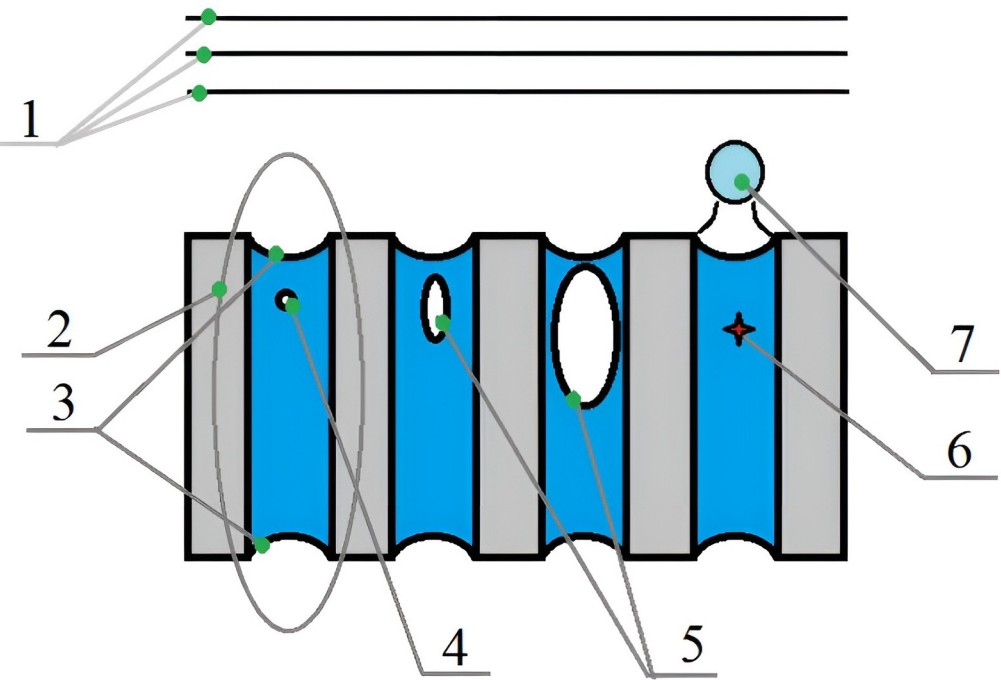

**Fig 1. Schematic drawing of the stages of energy conversion of ultrasonic vibrations when moisture is evacuated due to dispersion.** 1-ultrasonic vibrations in the air; 2-area for field computation in a separate capillary; 3-capillary menisci; 4-cavitation nucleus; 5-cylindrical cavitation pocket; 6-collapsed bubble; 7-dispersed drop.

The numerical submodel of the acoustic field is based on the equations of the solid phase elasticity theory for the particular case of sinusoidal dependence of the displacement on time:

$$-\omega^2 \rho \mathbf{u} = \frac{E}{2(1+\sigma)}\left(\Delta \mathbf{u} + \frac{1}{1-2\sigma}\nabla(div\mathbf{u})\right);$$

and propagation of vibrations in the liquid phase:

$$\Delta p + \frac{\omega^2}{c^2}p = 0;$$

where $\mathbf{u}$ is the displacement amplitude of the solid phase, m; $\omega$ is the circular frequency of ultrasonic vibrations, s$^{-1}$; $p$ is the pressure amplitude in the liquid phase, Pa.

The given equations are supplemented with the boundary conditions on the gas-liquid interface (solid phase surface—ambient gas):

$$\frac{E}{(1+\sigma)}\left(u_{ij}n_j + \delta_{ij}n_j div\mathbf{u}\right) = P_{g,A}\mathbf{n};$$

"liquid-solid" (capillary wall):

$$\frac{E}{(1+\sigma)}\left(u_{ij}n_j + \delta_{ij}n_j div\mathbf{u}\right) = p\mathbf{n},$$

where $P_{g,A}$ is the gas pressure amplitude, Pa.

In turn, on the surfaces of the capillary menisci, the pressure amplitude can be considered equal to zero due to the fact that the liquid wave-forming resistance is many times greater than

the wave-forming resistance of the gas phase. The solution of the given combined equations for 2related computational domains makes it possible to obtain the sound pressure distribution in a single capillary. Further, the energy conversion of liquid oscillations into the expansion of cavitation bubble is calculated with due regard to its deformation.

## Submodel of slow bubble growth due to the rectified diffusion

The slow bubble growth submodel is based on the analysis of the previously mentioned rectified diffusion phenomenon, when there is a unidirectional drift of the gas phase dissolved in the liquid into the gas cavity.

The unidirectional gas drift is performed due to the following diffusion mechanisms:

1. elastic compression and expansion of the bubble gas phase leads to the changes in gas concentration being proportional to the current concentration $Cg$ and inversely proportional to the current bubble radius (1/R). Moreover, the changes in concentration during compression and expansion are considered with a minus sign (the relative increase in the bubble radius entails a proportional relative decrease in the gas concentration according to the Mendeleev-Clapeyron equation);

2. diffusion of gas dissolved in the liquid through the interfacial layer (wall of the cavitation bubble) leads to the changes in concentration proportional to the difference in gas concentrations in the liquid and gas phases ($C_l$-$C_g$), and inversely proportional to the bubble radius R. This is due to the fact that the gas concentration in the bubble is determined as the gas weight divided by the bubble volume

$$C_g = \frac{m_g}{\frac{4}{3}\pi R^3}.$$

In turn, the total gas flow through the interfacial layer is linearly proportional to its area $J = 4\pi R^2 K \left( C_l - C_g \right)$. Since the gas flow through the interfacial layer is equal to the change in the total gas weight, then, accordingly, any change in concentration will be inversely proportional to the bubble radius ($R^2/R^3 = 1/R$).

With due regard to the described mechanisms, the availability of gas drift is due to the following:

1. during the half-period of discharge, the bubble is expanded, the gas concentration in the bubble is decreased; as a result, a positive difference in concentrations is developed near the phase boundary (bubble wall)—$C_l$-$C_g$> 0. Availability of a positive difference in concentrations leads to an increase in the total gas weight in the bubble $\Delta m_g = 4\pi R^2 K \left( C_l - C_g \right)\Delta t$. Finally, the total gas weight in the bubble is increased (the gas weight inside the bubble is changed to $\Delta m_g = 4\pi R^2 K \left( C_l - C_g \right)\Delta t$);

2. during the half-period of compression, the opposite process is implemented according to which the absolute rate of gas diffusion into liquid from the bubble during its compression is lower than the rate of gas diffusion from the bubble into liquid during its expansion. This is due to the fact that the bubble radius turns out to be larger at the beginning of the compression half-period than at the beginning of the expansion half-period. As it has been noted earlier, the absolute value of the diffusion rate is inversely proportional to the radius.

Thus, during the period of vibrations, there is a total increase in the gas weight inside the bubble.

The author of paper [23] have obtained an equation for the rectified diffusion of liquid-dissolved gas into the bubble:

$$\frac{dm}{dt} = \frac{\frac{8}{3}\pi DC_0 R_0 \left(\frac{p_m}{p_0}\right)^2}{\left(1 - \xi^2(R_0)\right)^2 + \xi^2(R_0)d^2(R_0)} - 4\pi D_r C_0 R_0; \qquad (1)$$

$$\xi(R_0) = \left(\frac{\omega}{\omega_0(R_0)}\right)^2; \qquad (2)$$

$$d(R_0) = \frac{1}{Q(R_0)} = \frac{\omega\eta}{K_c(R_0)}; \qquad (3)$$

where $m$ is the gas weight inside the bubble, kg; $D$ is the in-diffusion coefficient of the dissolved gas in the liquid, m²/s; $D_r$ is the back diffusion coefficient of the dissolved gas in the liquid, m²/s; $R_0$ is the radius of cavitation nucleus, m; pm is the sound pressure amplitude in the liquid, Pa; $p_0$ is the static pressure in the liquid, Pa; $\omega$ is the circular frequency of the acoustic impact, s⁻¹; $\omega_0$ is the resonant frequency of the cavitation bubble, s⁻¹, determined by the Minnaert formula $\omega_0 = \frac{1}{R_0}\sqrt{\frac{3\gamma}{\rho}\left(\frac{2\sigma}{R_0} + p_0\right)}$ (the for cavitation nuclei with a radius of less than 1 μm $\omega_0 \approx \frac{1}{R_0}\sqrt{\frac{3\gamma}{\rho}\frac{2\sigma}{R_0}}$); $K_c$ is the compressibility factor of the cavitation bubble, equal to $K_c = -\frac{1}{V}\frac{dV}{dp} = \frac{1}{\gamma p_0}$.

An analysis of the given equation has allowed the authors to obtain an analytical expression for the dependence of the bubble radius on time.

In view of the adiabatic relation $p / p^\gamma = const$, it is fair to assume that the gas density inside the bubble is constant throughout the entire rectified diffusion process. In addition, the value of $\xi(R_0)d(R_0) >> 1$ with a cavitation nucleus radius of less than 1 μm, makes it possible to allow the liquid-dissolved gas concentration to be close to the saturation concentration $C_0 \approx C_S$ according to [23, 24]. As a result of the assumptions made, the Eq (1) is transformed into a first-order ordinary differential equation (ODE) in relation to the cavitation nucleus radius

$$R_0{}^7 \frac{dR_0}{dt} = \frac{48 IDC_0 \sigma^2}{\rho^2 p_0{}^4 \omega^6 \eta^2}. \qquad (4)$$

The maximum size of the nucleus, achieved at the bubble nucleation stage τ, is determined according to the expression (4) obtained by the authors of the article:

$$R_0(\tau) = \sqrt[8]{R_0{}^8(0) + \frac{192 P_A{}^2 DC_0 \sigma^2}{\rho^3 c p_0{}^4 \omega^6 \eta^2}\tau}; \qquad (5)$$

where $P_A$ is the sound pressure amplitude, Pa.

The initial radius of the cavitation nucleus $R_0(0)$ is determined based on experimental data relating to the number and size distribution of bubbles. The papers [24, 25] provide information that the cavitation nucleus has a minimum size of 0.01 μm. The papers [26–28] present experimental data on the size distribution of the countable concentration of cavitation nuclei.

The performed approximation and analysis of such dependences allows to determine that with a decrease in the cavitation nucleus radius, the probability of its availability is increased

exponentially according to the inverse power function (in the range from 0.01 μm):

$$n_{bub} = \frac{C}{R^\alpha}.$$

It has made it possible to calculate the expected value of the cavitation bubble radius according to the following expression:

$$\langle R \rangle = \frac{\int\limits_{R_{\min}}^{R_{cap}} \frac{C}{R^\alpha} R dR}{n_{total}} \approx \frac{\dfrac{C R_{cap}^{2-\alpha}}{2-\alpha}}{n_{total}} = \frac{\dfrac{n_{bub0} R_0^{\alpha} R_{cap}^{2-\alpha}}{2-\alpha}}{n_{total}} \approx \frac{\dfrac{n_{bub0} R_0^{\alpha} R_{cap}^{2-\alpha}}{2-\alpha}}{n_{total}} \sim 1\ \mu m$$

The resulting value is taken as the initial size of the cavitation nucleus

$$R_0(0) = \langle R \rangle.$$

Thus, the proposed slow growth submodel makes it possible to calculate the dependence of the cavitation nucleus radius on time for the given parameters of acoustic field.

When the nucleus radius reaches a critical value [24, 25, 29] $R_0(\tau) > R_{crit}$, the bubble can begin rapid expansion within one vibration period and collapse instantly. The submodel for the rapid bubble expansion and collapse is described below.

## Submodel of rapid expansion and collapse of the cavitation bubble

The bubble expansion and collapse submodel is based on the general fluid dynamic equations:

$$\frac{1}{C^2} \frac{\partial H}{\partial t} + \frac{1}{C^2} (\nabla H, \nabla \varphi) + \Delta \varphi = 0; \tag{6}$$

$$H + \frac{|\nabla \varphi|^2}{2} + \frac{\partial \varphi}{\partial t} = 0; \tag{7}$$

the equations are supplemented with the boundary conditions on the bubble wall:

$$H|_s = \frac{p_0 + B}{\rho_0 \left(1 - \frac{1}{n}\right)} \left( \left( \frac{p|_s - 2\sigma K|_s + B}{p_0 + B} \right)^{1 - \frac{1}{n}} \right); \tag{8}$$

$$\nabla \varphi|_s = \frac{\partial \mathbf{r}}{\partial t}; \tag{9}$$

and on the capillary wall:

$$\left. \frac{\partial \varphi}{\partial \mathbf{n}} \right|_s = A_n;$$

where $H$ is the liquid enthalpy, m²/s²; φ is the liquid velocity potential, m²/s; $C$ is the speed of sound in the liquid, m/s; t is time, s; $S$ is the surface of the cavitation bubble wall; **r**(m) and K(m⁻¹) are the coordinate vector and the bubble wall curvature, respectively, m; $\rho_0$ is the equilibrium liquid density, kg/m³; $p_0$ is the static pressure in the liquid, Pa; $B$ (Pa) and $n$ are the constants in the Tait equation for the liquid state; $A_n$ is the amplitude of normal vibrations of the capillary wall, m.

To solve this equation, the numerical methods have been developed separately for the expansion stage and for the collapse stage.

The bubble expansion calculation method is based on the assumption that the fluid is incompressible in the vicinity of the bubble. In this case, the fluid velocity potential is represented as the following sum:

$$\varphi = \varphi_{bub} + \varphi_{osc},$$

where $\varphi_{osc} | \rho \frac{\partial \varphi_{osc}}{\partial t} = -\nabla p$; is the primary component of the acoustic field in the capillary ($p$ is the instantaneous value of the sound pressure in liquid located in the capillary, Pa); $\varphi_{bub}$ is perturbation caused by the motion of the bubble walls.

It is proved that the perturbation complies with the Laplace equation for the incompressible fluid:

$$\Delta \varphi_{bub} = 0;$$

and the boundary conditions on the bubble wall:

$$\frac{\partial \varphi_{bub}}{\partial t} = \frac{P_{osc}}{\rho};$$

where $P_{OSC}$ is the oscillatory component of pressure due to the acoustic field; $\frac{\partial \varphi_{bub}}{\partial \mathbf{n}} = \left( \frac{\partial \mathbf{r}}{\partial t}, \mathbf{n} \right)$ - the velocity potential gradient on the wall is equal to the velocity of the observed point of the wall, and the condition on the capillary wall:

$$\frac{\partial \varphi_{bub}}{\partial \mathbf{n}} = 0.$$

Thus, the problem of calculating the bubble expansion in the case of deformation is solved in the computational domain with the free (bubble wall) and fixed boundaries (capillary wall). In real practice, the capillary wall oscillates, however, these oscillations are considered in the primary component of the fluid velocity potential that satisfies the wall displacement boundary condition according to the submodel of the acoustic field in the solid and liquid phases. Hence, it follows that the boundary condition for the velocity potential perturbation $\frac{\partial \varphi_{bub}}{\partial \mathbf{n}} = 0$ is equivalent to the boundary immobility condition.

The initial position of the equivalent fixed boundary is determined by the capillary radius. The initial position of the free boundary is determined by the bubble radius with the assumption that the bubble is spherical upon completion of the slow growth phase. However, the slow growth model described above only allows to calculate the dependence of the cavitation nucleus radius on a given time, but does not provide any information on the cavitation nucleus radius at the boundary moment between slow growth and rapid expansion of the bubble.

To determine the moment after which the bubble will begin to collapse, as well as the radius of the latter, a direct enumeration of all times from the commencement of ultrasonic impact was performed. Then, at each radius, the Nolting-Neppiras dynamic equation of a spherical cavitation bubble were solved, in which $R(0) = R_0(\tau); \frac{dR}{dt}(0) = 0$; was used as an initial condition. Then, according to the collapse criterion (the bubble wall velocity reaches 1500 m/s), it was established whether the nucleus is capable of forming a shock wave $R_0(\tau)$.

Thus, for the numerical model of bubble expansion, the initial conditions are determined in full.

When solving the problem, the bubble and capillary wall are divided into the ring boundary elements, each of which, with the well-known cylindrical coordinates $(r_1; z_1)$ and $(r_2; z_2)$, is

specified parametrically (the problem is cylindrically symmetric):

$$\mathbf{r}(t, \varphi) = \begin{pmatrix} (r_1 + (r_2 - r_1)t)\cos \varphi \\ (r_1 + (r_2 - r_1)t)\sin \varphi \\ (z_1 + (z_2 - z_1)t) \end{pmatrix}; t = 0 \ldots 1; \varphi = 0 \ldots 2\pi.$$

The bubble wall position is explicitly determined by a set of pairs of cylindrical coordinates $(r_i; z_i)$ of the bubble wall points.

The general algorithm for the bubble deformation calculation at the expansion stage is as follows:

1. Solution of the Laplace equation $\Delta\varphi_{bub} = 0$ for a given shape and position of the bubble walls by the method of boundary integral equations:

$$\sum_{i=1}^{n} \int\limits_{S_{Ai}\cup S_{Bi}} E_{\frac{\mathbf{r}_i + \mathbf{r}_{i+1}}{2}} V_{ni}\partial S = \frac{\varphi\left(\frac{\mathbf{r}_i + \mathbf{r}_{i+1}}{2}\right)}{2} + \int\limits_{S_A \cup S_B} \frac{\partial E_{\frac{\mathbf{r}_i + \mathbf{r}_{i+1}}{2}}}{\partial \mathbf{n}} \varphi\partial S;$$

At the gas bubble boundary, the normal wall motion velocity component $V_n$ is an unknown quantity. On the capillary walls, the unknown quantity is the liquid velocity potential $\varphi$. The boundary integral equation at each time step is reduced to a system of linear algebraic equations.

$$\left\{A_{ij}\right\}_{i=1\ldots N, j=1\ldots N} \begin{Bmatrix} V_n^{(1)} \\ V_n^{(2)} \\ \ldots \\ V_n^{(N-1)} \\ V_n^{(N)} \\ \varphi_{cap}^{(1)} \\ \varphi_{cap}^{(2)} \\ \ldots \\ \varphi_{cap}^{(N-M-1)} \\ \varphi_{cap}^{(N-M)} \end{Bmatrix} = \left\{b_i\right\}_{i=1\ldots N}$$

Solution of the linear equation system at each time step leads to the following: $V_n^{(i)}$ are the normal velocity components of each bubble wall; $\varphi_{cap}^{(i)}$ is the liquid velocity potential on the capillary wall.

2. Depending on the determined normal components of the bubble velocity $V_n^{(i)}$, new positions of the boundary elements $\mathbf{r}(t, \phi) =$

$$\begin{pmatrix} \left(r_i + \Delta r_i + \left(r_{i+1} + \Delta r_{i+1} - r_i - \Delta r_i\right)t\right)\cos \phi \\ \left(r_i + \Delta r_i + \left(r_{i+1} + \Delta r_{i+1} - r_i - \Delta r_i\right)t\right)\sin \phi \\ \left(z_i + \Delta z_i + \left(z_{i+1} + \Delta z_{i+1} - z_i - \Delta z_i\right)t\right) \end{pmatrix}; t = 0 \ldots 1; \phi = 0 \ldots 2\pi \text{ are calculated}$$

based on the boundary condition $\frac{\partial \phi_{bub}}{\partial \mathbf{n}} = \left(\frac{\Delta \mathbf{r}_i}{\Delta t}, \mathbf{n}\right) = \left(\frac{\Delta \mathbf{r}_{i+1}}{\Delta t}, \mathbf{n}\right) = V_n^{(i)}$.

In this case, the capillary walls are considered fixed when calculating the bubble deformation, since their oscillation amplitude is rather small compared to the changes in the bubble radius for 1 period of ultrasonic vibrations.

3. Calculation of new velocity potential values on each boundary element based on the following condition:

$$\frac{\partial \varphi_{bub}}{\partial t} = \frac{P_{osc}}{\rho}.$$

4. Conditional check that at least one of the boundary elements of the bubble walls has reached the capillary wall. If so, then the algorithm is completed. It is assumed that the cavitation bubble does not collapse and no shock wave is generated.
Otherwise, it is necessary to proceed to step 5.

5. Conditional check that there is a decrease in the bubble volume compared to the previous time step. If so, then the algorithm is completed and the bubble collapse calculation shall be commenced.
Otherwise, it is necessary to proceed to step 6.

6. Time increment by $\Delta t$.

7. It is necessary to proceed to step 1.

After determination of the cavitation bubble shape using a numerical expansion model, the bubble collapse is calculated (for the cases where the bubble does not reach the capillary walls).

To calculate the collapse, the asymptotic decomposition was used represented by the following expressions:

$$\frac{\partial H_n}{\partial t} + \sum_{m=0}^{n} \frac{\partial H_m}{\partial r} \frac{\partial \varphi_{n-m}}{\partial r} - \frac{1}{r^2} \sum_{m=0}^{n} m(n-m) H_n(r,t) \varphi_{n-m}(r,t) +$$

$$+ \frac{1}{r^2} \sum_{m=0}^{n} (m+1)(n+1-m) H_{m+1}(r,t) \varphi_{n+1-m}(r,t) +$$

$$+ \left( \frac{1}{r^2} \left[ (2n-n^2)\varphi_n + \left((n+2)^2 - n - 2\right) \varphi_{n+2}(r,t) \right] + \frac{\partial^2 \varphi_n}{\partial r^2} + \frac{2}{r} \frac{\partial \varphi_n}{\partial r} - \frac{n \varphi_n}{r^2} \right) C_\infty^2 +$$

$$+ \sum_{m=0}^{n} \left( \frac{1}{r^2} \left[ (2m-m^2)\varphi_n + \left((m+2)^2 - m - 2\right) \varphi_{m+2}(r,t) \right] \right) (N-1) H_{n-m} +$$

$$+ \sum_{m=0}^{n} \left( \frac{\partial^2 \varphi_m}{\partial r^2} + \frac{2}{r} \frac{\partial \varphi_m}{\partial r} - \frac{m \varphi_m}{r^2} \right) (N-1) H_{n-m} = 0$$

The boundary conditions represent decomposition of each component of the liquid enthalpy and the velocity potential into a series in terms of harmonics of the deformed bubble surface.

Thus, the submodel makes it possible to calculate the time dependence of the bubble radius and pressure of the shock wave generated during the bubble collapsing process.

## Submodel of the liquid drop generation

The drop generation is determined by the kinetic energy that the capillary liquid acquires under the impact of shock waves. Since cavitation in a single capillary had been analyzed earlier, the volume of liquid splashed out from a single capillary was initially calculated.

The calculation algorithm for the spilled liquid volume is as follows:

1. Calculation of the dependence of the shock wave pressure $P(r, z, t)$ on the coordinates and time based on the acoustics equation (on the bubble surface, the shock wave pressure distribution determined at the previous stage is taken as the boundary condition).

2. Calculation of the distribution of velocities acquired by the liquid at each of its points, according to the following expression

$$\mathbf{v} = -\int \frac{\nabla P}{\rho} dt.$$

3. Calculation of the maximum volume of liquid splashed out of the capillary, according to the energy conservation law (the jet is approximated by a cylindrical shape, the jet diameter is equal to the capillary diameter).

$$\int_V \frac{\rho v^2}{2} dV = 2\sigma \pi a l - \sigma K_0 a^2;$$

$$V_{jet} = \pi a^2 l;$$

$$V_{jet}[v, a] = \frac{\sigma K_0 a^2 + \int_V \frac{\rho v^2}{2} dV}{2\sigma} a;$$

where a is the capillary radius, m; $\sigma K_0 a^2$ is a surface-tension energy of the capillary meniscus in the unperturbed state, J.

Further, the moisture evacuation efficiency [m/s] per unit surface area of a capillary-porous body is calculated at a given sound pressure:

$$\Pi_{per\ unit}(P) = K_{flow} \int_0^\infty I(P, a) \frac{V_{jet}(P,a)}{T} n_{bub}(P) V_{cap}(a) f(a) da;$$

where $P$ is the sound pressure amplitude in the adjacent gas phase, Pa; $n_{bub}$ is the concentration of cavitation nuclei in the liquid, m$^{-3}$; $V_{cap}(a)$ is the volume of an individual capillary with a radius a, m$^3$; $I(P,a)$ is an indicator function equal to 0 if, at a given sound pressure $P$ and capillary radius a, the bubble reaches the capillary walls or there is no cavitation bubble with collapse at all (only the rectified diffusion phenomenon is remained), and equal to 1, if otherwise; $K_{flow}$ is coefficient taking into account flow of moisture from sample interior and which depends on acoustic field in interior (non-dimensional variable).

To calculate the moisture evacuation efficiency due to atomization from the entire surface of the dried sample, the sound pressure distribution in the gas phase is calculated based on the Helmholtz equation:

$$\Delta P_A + k^2 P_A = 0;$$

where $P_A$ is the sound pressure amplitude, Pa; in the boundary conditions of vibration reflection in the sample surface and at a given sound pressure amplitude.

Further, the total efficiency of moisture evacuation from the material (drying) is calculated according to the following expression:

$$\Pi_{total} = K_{flow} \int_S \Pi_{per\ unit}(P_A(\mathbf{r})) dS(\mathbf{r}).$$

Thus, all the mathematical expressions required to identify the ultrasonic exposure modes and conditions, under which the moisture evacuation from the dried material is performed in a dripping form without any conversion to steam, have been obtained.

## Materials and methods

### Design of the experimental setup

To experimentally determine the moisture evacuation efficiency in a droplet form and confirm the theoretically identified modes of ultrasonic exposure, a stand for experimental studies of drying was developed and manufactured (Fig 2) [30].

The stand was made by upgrading the SHD25 drying plant produced by"Shini Plastics Technologies" (China). To provide ultrasonic exposure, a disc ultrasonic emitter 3 was installed in the cover 2 of the body of the drying chamber 1. A reflecting disk 4 was located in the lower part of the dryer. Due to the adjusting screw 5, the reflecting disk could be moved along the axis of the drying chamber, thereby ensuring the development of standing acoustic waves in the drying chamber. Warm air entered the drying chamber through the inlet 6. The air was heated by an electric heater 7. The heated air was exhausted by the fan 8. The warm air was exhausted from the dryer through the outlet pipe 9. The air 10 was moved parallel to the axis of the cylindrical drying chamber towards the disc emitter. The pallet with the material to be dried 11 was located in the drying chamber, at some distance from the disc emitter. The glass slides with the immersion medium 12 could be placed near the pallet and in the outlet pipe to trap the liquid drops dispersed from the material to be dried by ultrasonic impact.

The temperature of the drying air $T$ was set in the range of 30–120°C with an accuracy of ±1°C. The air flow velocity S in the dryer housing was controlled by a damper in the range from 0.5 to 1 m/s (the measurements were made with a UNI-T UT363S digital anemometer). The air humidity in the room was measured with a Testo 625 thermohygrometer. The electric power consumption of the heater and fan was 950 W at a temperature of $T = 40$°C; 1480 W at $T = 60$°C; and 2120 W at $T = 80$°C.

The appearance of the stand is shown in the photo (Fig 3).

### Design of the ultrasonic emitter

A bending oscillating disc emitter with a diameter of 205 mm was used as a source of ultrasonic exposure. A piezoelectric oscillatory system based on a Langevin transducer was used to excite oscillations of the disc emitter [31]. The design of a disk emitter assembled with a piezoelectric oscillatory system is shown in Fig 4.

An electrical signal of ultrasonic frequency is applied through the electrodes to the piezoceramic disc elements 3 that convert it into the mechanical vibrations. The acoustic coupling within the oscillatory system is provided due to the fact that the piezoceramic elements are clamped between the emitting concentrator 1 and the reflective pad 4 with a force many times greater than the alternating force produced by the piezoelectric elements. The emitting concentrator provides an increase in the amplitude of vibrations that are transmitted to the disc emitter 2.

Since the disc emitter thickness is much less than the wavelength, when the disc emitter is excited, the bending vibrations with various vibration modes, called the modes of oscillations,

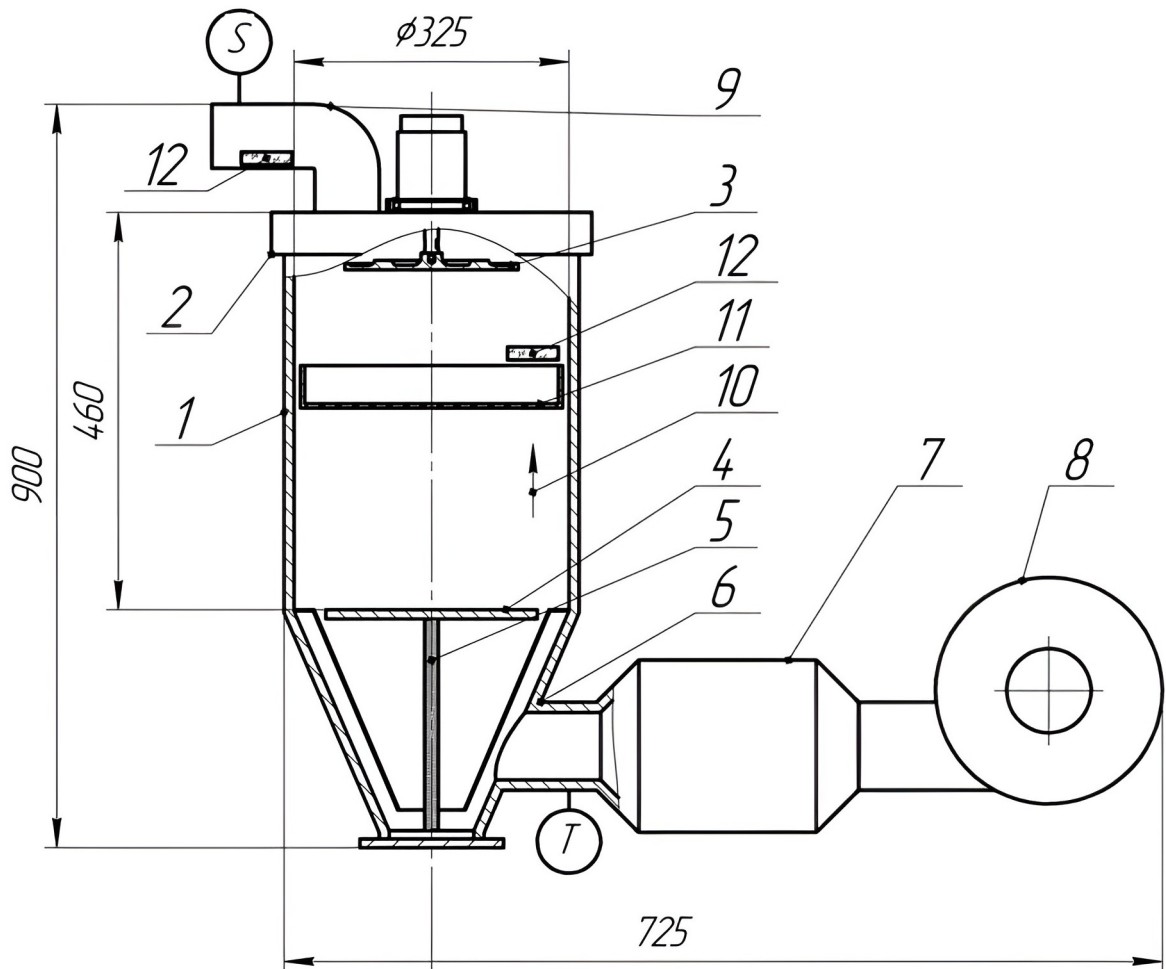

**Fig 2. Schematic block diagram of the test stand.** 1-body of the drying stand; 2-cover of the drying stand; 3-disc emitter; 4-reflecting disk; 5-adjusting screw; 6-heated air supply pipe; 7-electric heater; 8-fan; 9-heated air exhaust pipe; 10-direction of the heated air; 11-pallet with the dried material; 12-glass slides with the immersion medium; S-air flow velocity sensor; T-temperature sensor.

can be generated. In this case, the adjacent annular regions of the disc emit oscillations in the opposite phases. That is, some areas are oscillated in phase with the oscillatory system, while others are oscillated in an out of phase way. Accordingly, between the adjacent areas oscillating out of phase, there are the areas which oscillation amplitude is equal to zero. Thus, the adjacent annular regions emit oscillations into the gaseous medium in the opposite phases while leading to the fact that there is a mutual compensation of emission at some distance from the disc.

In order to reduce this negative effect, the disc is made in a stepwise variable fashion with an increase in the disc thickness in the areas that emit the antiphased oscillations.

The appearance of the emitter with an electronic generator is shown in Fig 5.

Table 1 demonstrates the technical specifications of the ultrasonic device.

## Simulation of sound pressure level distribution

To determine the distribution of vibrations, distribution of the sound pressure level generated by a disc emitter was simulated (Fig 6).

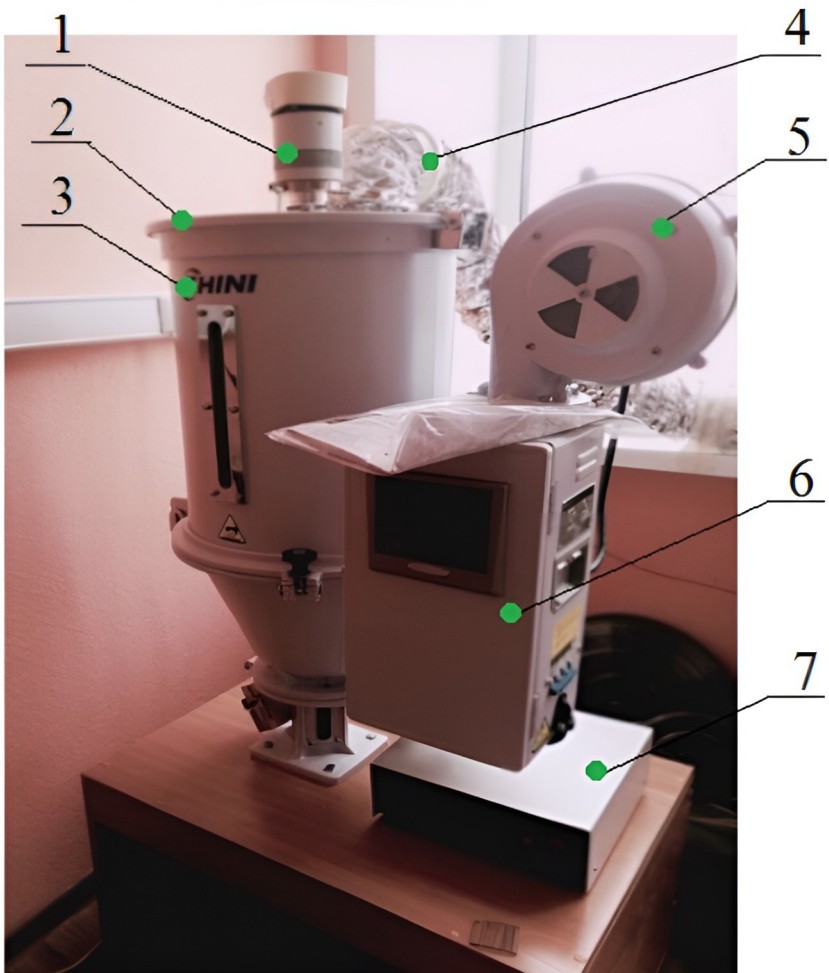

**Fig 3. Photo of the test stand.** 1-housing of the disc emitter; 2-cover of the drying stand; 3-body of the drying stand; 4-air outlet pipe; 5-fan; 6-control cabinet; 7-ultrasonic generator.

The simulation of acoustic field generated by the emitter used was performed by the final element analysis in the ANSYS system by using the harmonic acoustic analysis module (Harmonic Acoustics).

To compare the distributions of sound pressure level generated by an ultrasonic emitter in the presence and absence of the reflecting boundaries (walls of the drying chamber), the ultrasonic field was simulated for two cases: an ultrasonic emitter installed in an infinite area without any reflecting boundaries and in a drying chamber.

For the case with the emitter location in the drying chamber, a three-dimensional model of the inner cavity of the drying chamber was developed with the cut-out region of the disc emitter. In this case, the surfaces of the inner chamber walls were set as the reflecting boundaries, the absorption of which did not exceed 15% being consistent with the results of experimental studies performed earlier by the authors.

In the case of absence of the reflecting boundaries, a spherical region was specified, the outer boundary of which was taken as an absorbing one (emitting in the outward direction). When setting up the acoustic analysis, the frequency and distribution of vibrational velocities

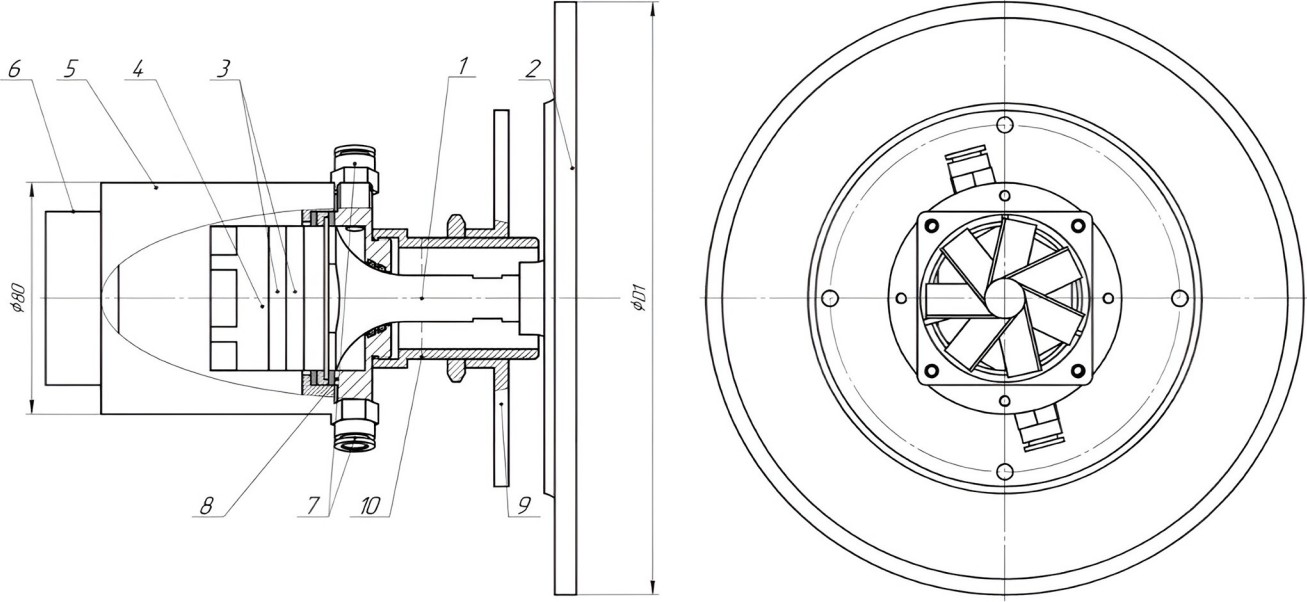

**Fig 4. Design sketch of an ultrasonic piezoelectric transducer with a disc-shaped emitter.** 1-emitting concentrator of the piezoelectric transducer; 2-bending oscillating disc; 3-piezoceramic rings; 4-reflective pad; 5-body; 6-fan; 7-fittings for the coolant supply/exhaust; 8-heat exchanger; 9-flange; 10-threaded flange; D1-disc diameter.

developed by the disc emitter were set being obtained on the basis of the measurement results of its vibration amplitudes.

When setting the parameters of a three-dimensional finite element mesh, a tetrahedral finite element type was applied. A tetrahedral finite element mesh approximates the objects of compound shape rather well and provides satisfactory simulation results for the objects of arbitrary shape, the specific geometric dimensions of which are comparable to each other in terms of three dimensions (length, width, height).

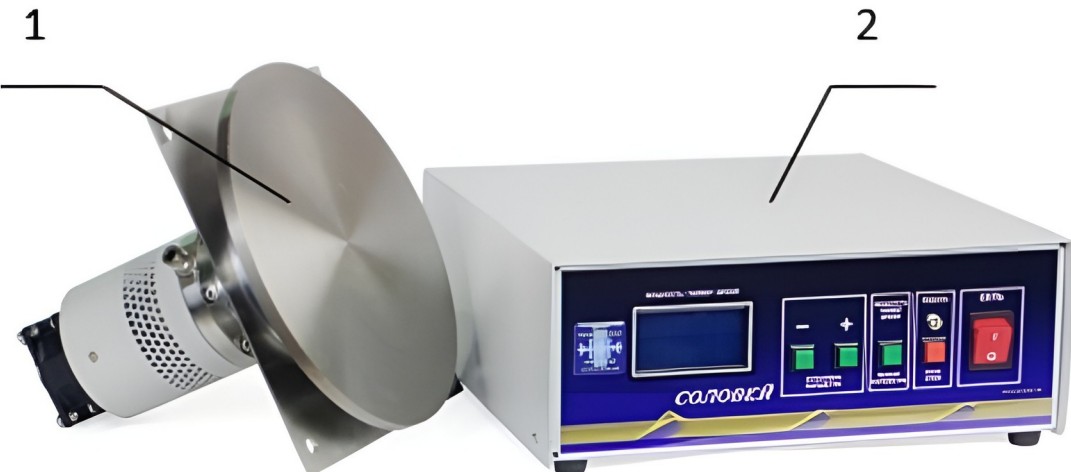

**Fig 5. Appearance of the Solovey ultrasound device for gas media.** 1-ultrasonic oscillatory system with a disc emitter; 2-electronic generator.

**Table 1. Technical specifications of the ultrasound device.**

| Name of the parameter | Value |
|---|---|
| Power supply from the AC mains with the voltage, V | 220±22 |
| Maximum power consumption, not more than, VA | 300 |
| Power control range, % | 40–100 |
| Mechanical oscillation frequency of the emitter, kHz | 22.0±0,1 |
| Maximum sound pressure level (within 1 m), dB, not less than | 150 |
| Overall dimensions of the electronic control unit, mm | 400x280x110 |
| Overall dimensions of the ultrasound oscillation system, mm | Ø205x270 |
| Emitting surface diameter, mm | 205 |
| Cooling system | Forced air |

When performing the acoustic analysis, the convergence of numerical results was also analyzed. The simulation result was considered satisfactory if it corresponded to the finite element model with a minimum number of finite elements. An increase in this number of finite elements led to the changes in the base values of calculated parameters (for example, the maximum sound pressure level) by no more than 0.2–0.5%. Thus, it was found that the linear dimension of the tetrahedral mesh element was 3 mm.

The simulation procedure made it possible to determine that a standing wave with the regular arrangement of oscillation maxima was generated throughout the height of the drying chamber in its central vertical section. The average sound pressure level in the drying chamber was 155 dB, and the maximum level was equal to 172 dB.

In turn, the distribution of sound pressure levels in the central horizontal section of the drying chamber also had a regular pattern, but in the form of circles. The number of peaks in the sound pressure level corresponded to the number of half-waves of the disc bending-oscillating emitter (Fig 6).

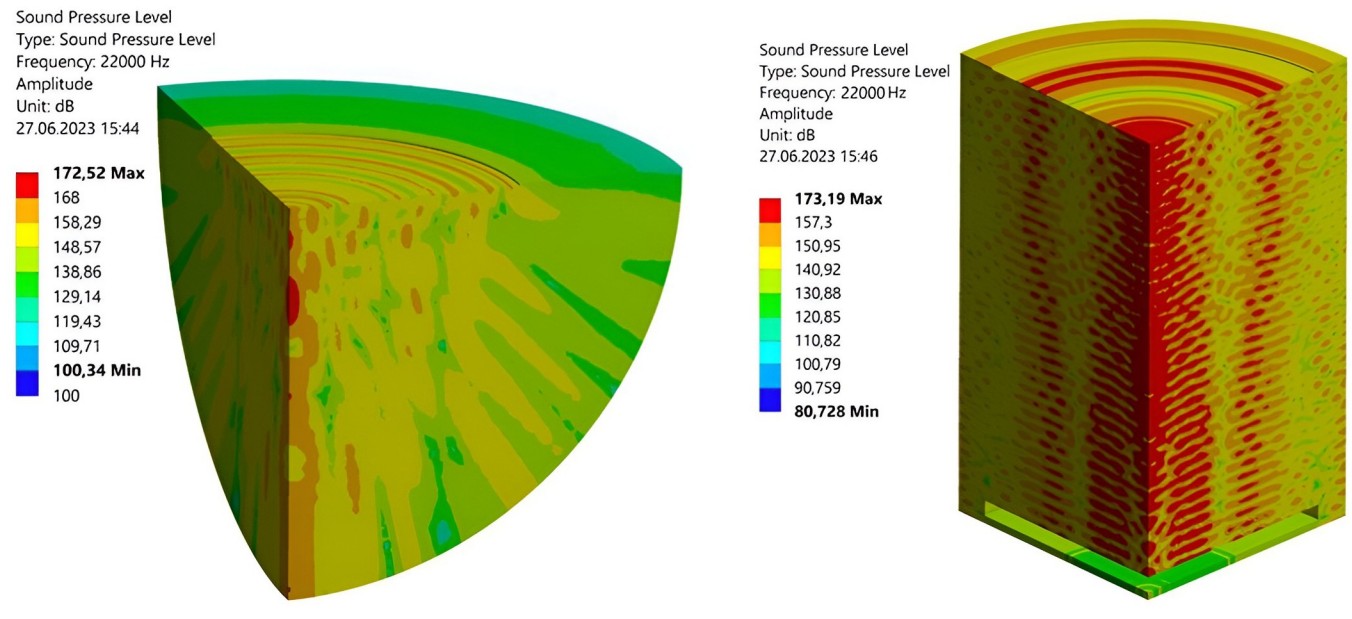

**Fig 6. Distribution of the acoustic field.** (a) in the open space; (b) in the drying chamber.

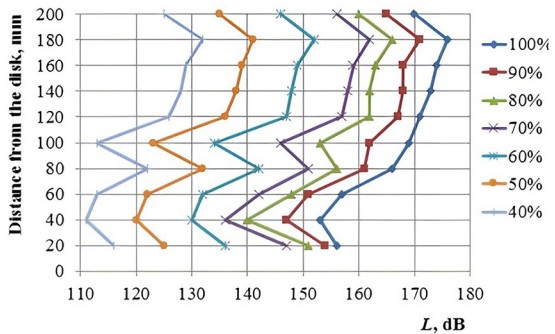 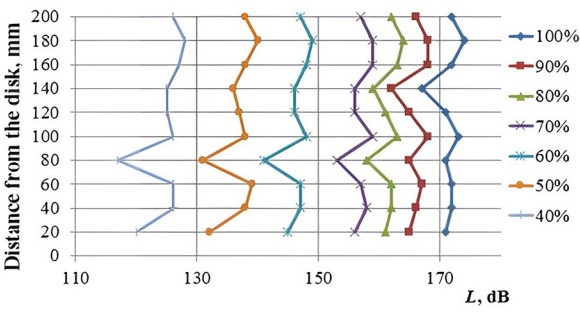

**Fig 7. Distribution of sound pressure levels *L* depending on the distance from the emitter at different values of the output power settings of the electronic generator (%). (a)** on the acoustic axis of the disc; **(b)** at a distance of 70 mm from the disc axis.

Thus, it has been established that the acoustic field structure developed in the drying chamber by the disc emitter has a regular structure both in the longitudinal direction (in the direction of the emitter's acoustic axis) and in the transverse direction (along the emitter's radius).

## Verification of simulation results

To confirm the calculation results, we measured the sound pressure level generated by the ultrasonic emitter in the dryer. The measurements were taken along the axis of the disc emitter and at a radial distance of 70 mm (Fig 7, S1 File). The measurements were taken using an Eko-fizika-110A/Inzhener-110A noise and vibration metering device with a VMK-401 microphone at various setting (power) levels of the electronic generator [32].

It follows from the measurement results that the sound pressure level distribution in the body is not uniform throughout the height of the drying stand body. The maximum sound pressure level along the central axis of the dryer cylinder and at a radius of 70 mm from it is reached at a distance of 180 mm from the disc emitter. The values of sound pressure levels at a distance of 180 mm from the disc are given in Table 2.

It follows from the table that at a distance of 180 mm (a multiple of the wavelength of ultrasonic vibrations in the air) from the disc emitter, the sound pressure levels can be obtained in increments of 5 dB at various generator output power settings. In this case, the spread does not exceed ±2 dB. The obtained results of sound pressure levels make it possible to perform the ultrasonic drying tests in a wide range of sound pressure levels from 130 to 175 dB.

## Control of moisture evacuation in a droplet form

To confirm the physical mechanism of moisture evacuation from the material being dried in a droplet form by atomization, the entrapping method for the droplets generated by an

**Table 2. Sound pressure levels at a distance of 180 mm from the disc emitter.**

| Distance from the disc axis, mm | Setting level of the electronic generator from the maximum power, % / power consumption, W | | | | | | |
|---|---|---|---|---|---|---|---|
| | 40/50 | 50/60 | 60/70 | 70/100 | 80/150 | 90/170 | 100/200 |
| | Sound pressure level *L*, dB | | | | | | |
| 0 | 132 | 141 | 152 | 162 | 166 | 171 | 176 |
| 70 | 128 | 140 | 149 | 159 | 164 | 168 | 174 |
| **Average value** | **130** | **140,5** | **150,5** | **160,5** | **165** | **169,5** | **175** |

immersion medium was applied. The glass slides with the immersion medium were installed both in the immediate vicinity of the material to be dried, and in the outlet pipe of the drying chamber (Fig 2). Subsequently, the dimensions of the trapped drops and their size distribution were determined by assessing the size and number of trapped drops using a Mikmed-6 microscope manufactured by Lomo (Russia) [33].

## Materials

To confirm the possibility and efficiency of moisture evacuation in a droplet form from the capillary porous materials and to verify the theoretically obtained results, an agglomerated cork plug produced by Morell was selected as the material to be dried. The selection of material for research was due to the fact that the cork has a developed system of three-dimensional hexagonal pores with the size of about 20 μm [34] (Fig 8). This material retains its original dimensions during the drying process and is not subject to deformations. The cork can be repeatedly soaked with moisture to obtain a given initial moisture content that has provided for the reproducibility of numerous tests.

The cork was cut into the cube-shaped specimens with the edge lengths of 10 mm, 15 mm, and 20 mm. Each group of wet samples was given the same weight of 100 g ± 0.1 g.

The cork was impregnated by its soaking in the settled main water. The cork plug samples placed in water were exposed to the ultrasonic vibrations for 40 minutes that ensured uniform impregnation of the samples [35]. The moisture content was controlled by the gravimetric method. The initial moisture content of the cork plug $W_n$ in all tests was 0.192 kg/kg. The drying process was continued until the material reached a moisture content of 0.05 kg/kg.

Since it is known that the ultrasonic exposure is most appropriate to use for drying the food products [36–38], then the beetroot is used as a dried material for practical testing of the ultrasonic drying process.

The fresh red beet (Beta vulgaris L.), "Flat of Egypt" variety used in this study was purchased in the Metro supermarket. Prior to drying, the vegetables were stored in a refrigerator at a

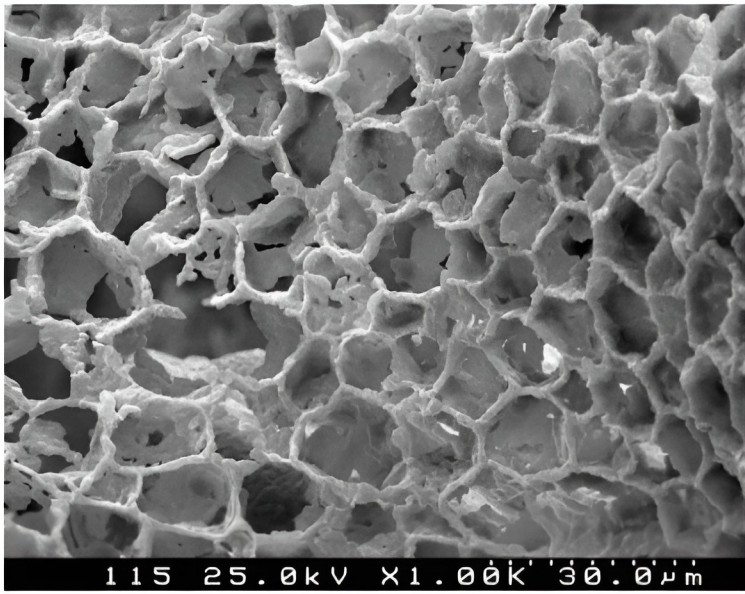

**Fig 8. Photomicrograph of the cork plug.**

temperature of 4˚C, after which they were washed and cleaned by hand. Then the beets were cut into the cubes with the dimensions of 15x15x15 mm. The initial weight of fresh beet samples for each test was 250±0.1 g.

The average initial moisture content of beet WB in all tests was 6.5 kg/kg. The drying was continued until the samples reached a final moisture content of 1.0 kg/kg.

## Energy consumption for the drying process

To determine the energy efficiency of ultrasonic drying *EE*, we used the method to compare the amount of electric energy consumed for the convection drying and ultrasonic drying at the constant parameters of convection drying and the material being dried:

$$EE = (1 - \frac{E_{US}}{E_c}) \cdot 100\%,$$

where $E_c$ is the amount of electric energy consumed for the convection drying, kW·h; $E_{US}$ is the amount of electric energy consumed for the ultrasonic drying, kW·h. The amount of electric energy consumption was measured by a Zhurui electric energy meter PR10.

## Experimental method

During the experiments, the following parameters were kept constant: the flow rate of the drying air was 0.5 m/s for the cork plug and 1.0 m/s for the red beet, the air humidity in the room was 40%, while the drying air temperature was 30˚C during the cork plug drying and 50˚C during the drying of red beets.

The moisture content was determined by weighing the samples of the dried material at the regular intervals on a Pocket scale MH-300 with an accuracy of ±0.01 g. The moisture content of the material during the drying process was determined by the following formula:

$$W = \frac{m_l}{m_0},$$

where $m_l$ is the liquid weight, kg; $m_o$ is the weight of dry material, kg.

The drying rate was determined by the numerical differentiation method:

$$\frac{dW}{d\tau} = -\frac{W_{i+1} - W_i}{\tau_{i+1} - \tau_i},$$

where $W_i$ is the moisture content of the material (kg/kg) at time $\tau_i$ (s); $W_{i+1}$ is the moisture content of the material (kg/kg) at time $\tau_{i+1}$ (s).

In the studies performed, various methods of sample placement during the tests were used.

To confirm the possibility and efficiency of moisture evacuation from the material in a droplet form without any transformation into the steam, the cork plug samples were placed on a wire-mesh pallet, in a place where the average value of the sound pressure level is the highest.

To confirm the possible moisture evacuation from the dried material in a droplet form without any transformation into the steam, the cork plug samples of all sizes (10 mm, 15 mm and 20 mm) were placed on a wire-mesh pallet. The total weight of the wet samples was 100 g. The samples with different sizes were placed randomly on the pallet.

During the test aimed at determining the optimal size of dried samples, the cubes with one of the sizes were placed on a pallet as a part of each test. The experiments were repeated for the cork cubes of all sizes, ranging from the lowest to the largest ones.

When confirming the practical applicability of the proposed moisture evacuation mechanism to the food capillary-porous products, the dried samples were placed on several pallets

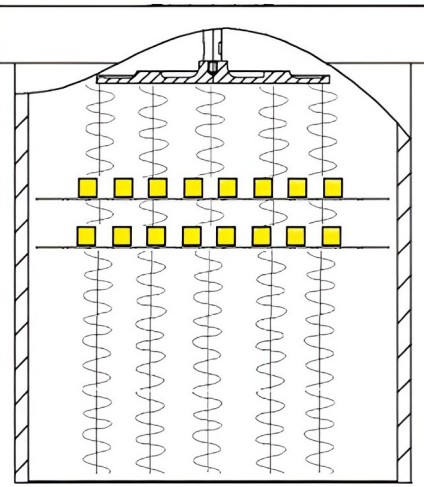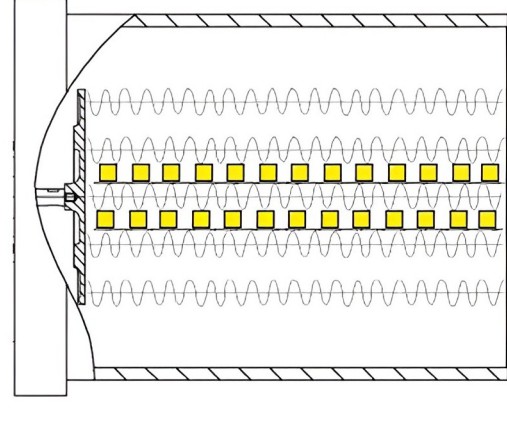

**Fig 9. Propagation of ultrasonic vibrations in relation to the dried material. (a)** location perpendicular to the acoustic axis of the emitter; **(b)** location parallel to the acoustic axis of the emitter.

simultaneously. Moreover, the pallets were direvted in two ways: 1) perpendicular to the acoustic axis of the emitter and the drying agent movement; 2) parallel to the acoustic axis of the emitter and the direction of air flows (Fig 9).

The samples of red beets were placed on two lattice pallets at a distance of 180 mm and 150 mm from the ultrasonic disc emitter with the direction of exposure to the ultrasonic vibrations and air flow perpendicular to the pallet surface. In the case of the parallel exposure to the ultrasonic vibrations and air flow, the pallets were placed near the acoustic axis of the emitter at a distance of 3 cm from each other (Fig 9). The sound pressure level in various tests was 165±1 dB.

## Results and discussion

### Justification of the moisture evacuation mechanism in the droplet form

The theoretical calculations of an acoustic field generation in the liquid and solid phases, as well as the generation of cavitation bubbles, have made it possible to confirm the possible implementation of the moisture evacuation mechanism in a droplet form. They are based on determination of the acoustic field parameters and allow to obtain the sound pressure distribution in a single capillary shown in Fig 10. All calculations were performed at following input data: frequency of ultrasonic oscillations– 22 kHz; modal diameter of capillary– 10 μm; RMS deviation of diameter of capillary– 8 μm; sound speed in air– 343 m/s; sound speed in sample– 660 m/s; density of air– 1.22 kg/m3; density of sample– 400 kg/m$^3$; poisson ratio of sample– 0.3; Young's modulus of sample– $1.4 \cdot 10^{10}$ Pa; ultrasonic attenuation coefficient in sample– 20 m$^{-1}$.

The given sound pressure distribution was obtained at an amplitude of normal vibrations of the capillary walls of 0.2 μm with a capillary diameter of 10 μm. From this point on, water is taken as the liquid phase under normal conditions being the evacuated moisture. As the obtained distribution has showed, at a given displacement amplitude, the sound pressure amplitude reaches $1.5 \cdot 10^5$ Pa. The specified sound pressure determines the possible cavitation.

The indicated sound pressure amplitude at a small displacement amplitude of the capillary walls is due to the liquid inertia in the capillary that contributes not only to the vertical liquid

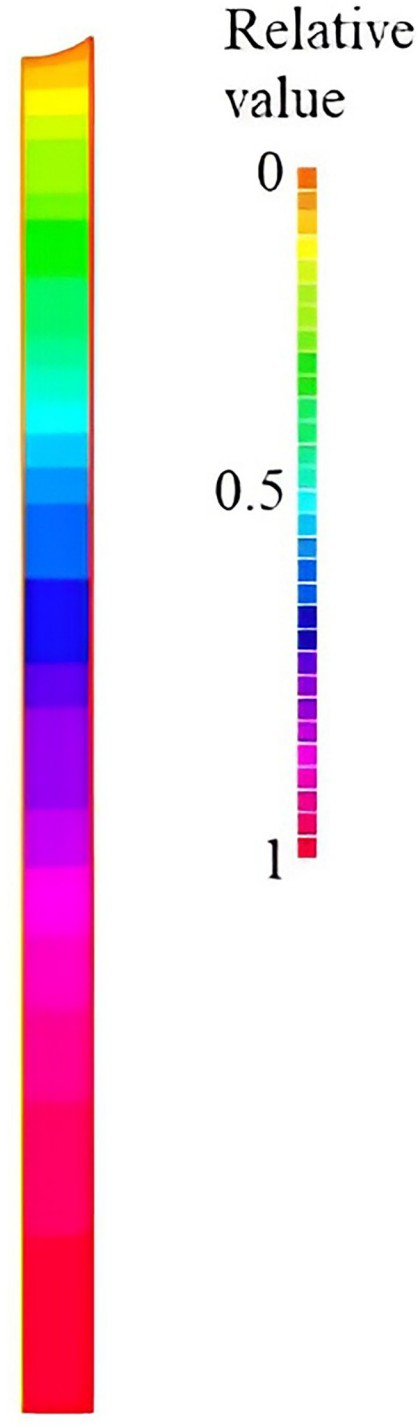

**Fig 10. Sound pressure distribution in the liquid volume located inside the capillary (relative value 1 corresponds to 152000 Pa).**

displacement, but also to its partial volumetric compression (the liquid mass prevents its vertical displacement, and, therefore, the liquid is partially subject to the transverse compression). In the limiting case, when the lateral boundaries of the liquid are not free (the capillary menisci), but are absolutely fixed (the capillary is closed at the ends) or the liquid volume has an infinite length, the pressure amplitude (at a similar amplitude of normal displacements of the capillary walls) is $\frac{A}{R}\rho c^2 = 9 \cdot 10^7\ Pa$, namely the pressure that is more than 80 times the cavitation occurrence threshold in water.

At the determined sound pressure, the expansion process of the cavitation bubble was calculated with due regard to the deformation using the cavitation bubble expansion and collapse submodel (Fig 11).

Based on the obtained dimensions and shape of the cavitation bubble, the generated shock wave parameters were calculated. It was found that the shock wave pressure amplitude could reach $10^7$ Pa or more during a period of time of 0.1 μs. The shock wave profiles in the capillary with a diameter of 10 μm are shown in Fig 12.

Further, the droplet generation calculations were performed. The first stage of drop generation calculations consisted in determination of the specific atomization productivity from a unit surface near a small area (where the sound pressure does not depend on the coordinates). The dependence of the moisture evacuation efficiency on the sound pressure level in the

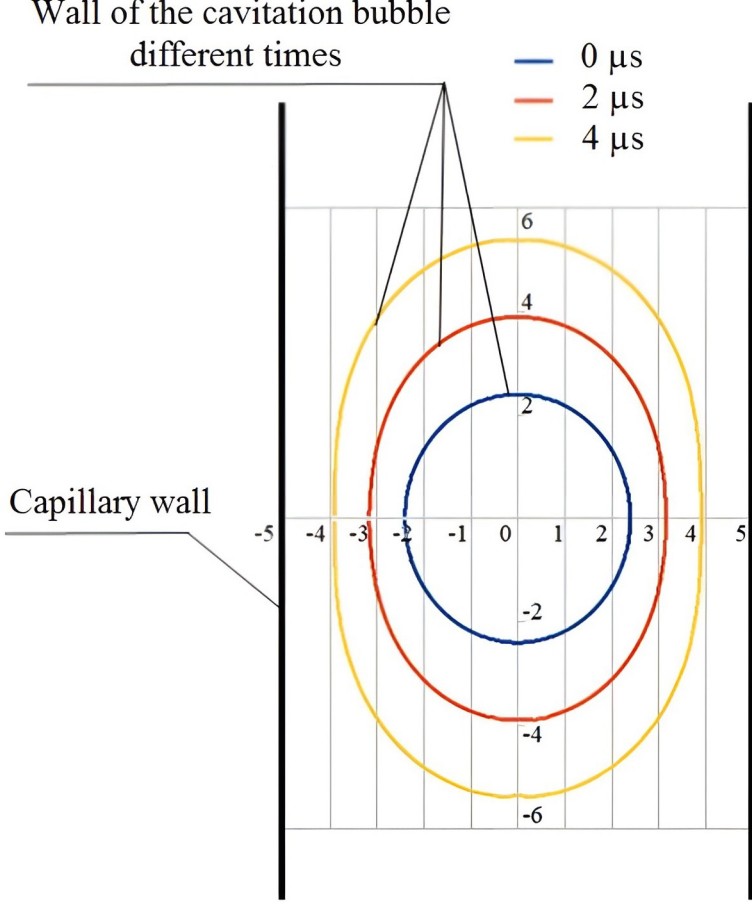

**Fig 11. Deformation of the cavitation bubble over time at the expansion stage (the axial coordinates are given in microns, the capillary diameter is 10 microns).**

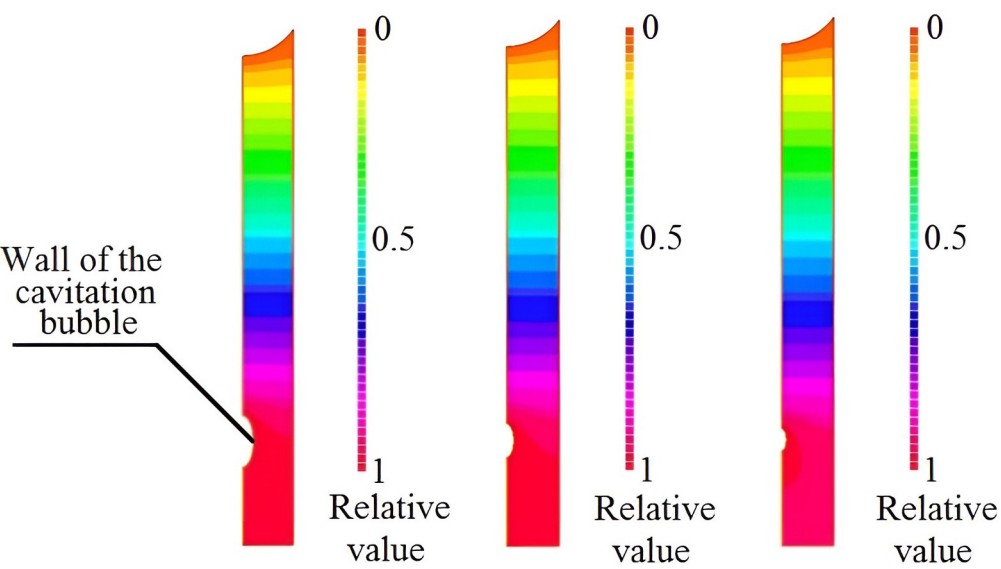

**Fig 12. Pressure distributions of the shock wave in the case of the cavitation bubble collapse in a capillary with a diameter of 10 microns. (a)** sound pressure level in gas, 150 dB (maximum shock wave pressure is 75.5 MPa, that corresponds relative value 1); **(b)** sound pressure level in gas, 160 dB (maximum shock wave pressure is 482 MPa, that corresponds relative value 1); **(c)** sound pressure level in gas, 170 dB (maximum shock wave pressure is 3076 MPa, that corresponds relative value 1).

adjacent gas phase is shown in Fig 13. The dependence is obtained with due regard to the poly-disperse capillary size distribution.

Analysis of the dependence obtained allows to conditionally distinguish 3 areas of changes in the drying efficiency depending on the sound pressure level:

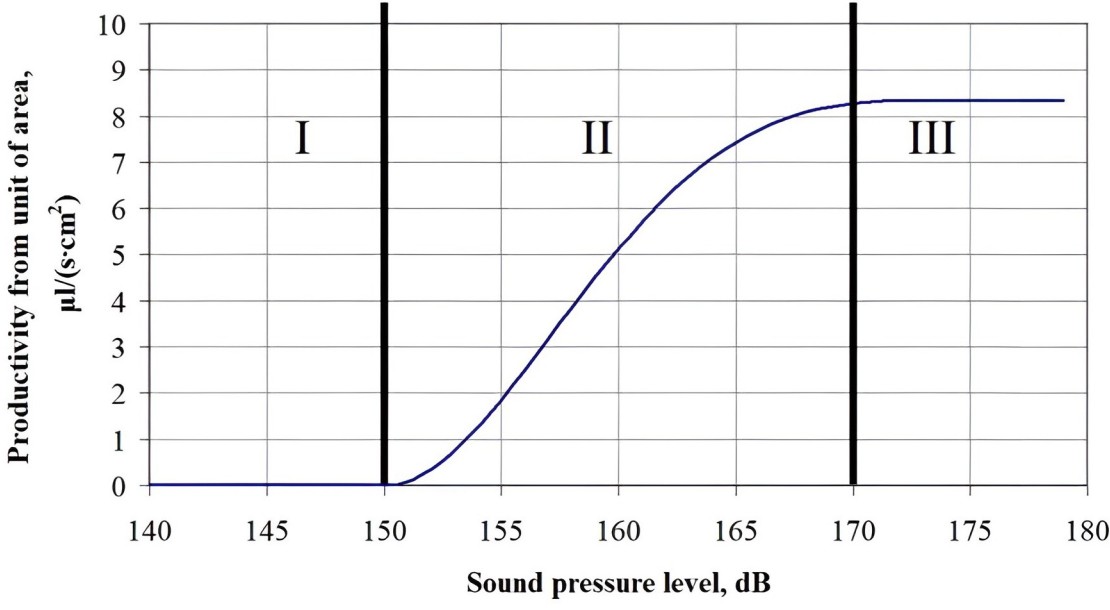

**Fig 13. Dependence of the moisture dispersion efficiency from a volume of 1 cm³ on the sound pressure level.** Range I–no cavitation and, therefore, no dispersion; Range II–intense development of cavitation with an increase in the sound pressure level (optimal range); Range III–saturation of capillaries with the gas bubbles and cessation of bubble collapse in a part of capillaries.

1. In the first section, with an increase in ultrasonic exposure to the sound pressure level of less than 150 dB, a slight growth of the dispersion (moisture evacuation) efficiency is noted with an increase in the sound pressure level;

2. In the second section, when the sound pressure level of ultrasonic exposure is changed from 150 to 170 dB, a sharp increase in the dispersion efficiency is observed with increasing sound pressure level (the growth function shape is close to the linear one);

3. In the third section of the obtained dependence, at a sound pressure level of more than 170 dB, a cessation in the dispersion efficiency growth is observed.

It is obvious that an increase in the sound pressure during the drying process requires certain energy costs. Thus, such a slight increase in the efficiency (performance) in the third section of the analyzed dependence that occurs with a significant increase in the power costs for the ultrasonic vibrations, requires a separate analysis. Therefore, the possible practical implementation of drying process under ultrasonic exposure with a sound pressure level of more than 170 dB raises significant doubts.

It follows from the dependences obtained that the drying process under ultrasonic exposure with a pressure level of up to 150 dB in the first section is not efficient, since the liquid evacuation by dispersion does not occur and, therefore, the process almost does not differ from the convection drying.

In the second section, moisture evacuation without a phase transformation occurs more intensively, while reaching a maximum at a pressure level of 165–170 dB.

As already noted, in the third section there is a cessation in the dispersion efficiency growth that is related to a decrease in the moisture diffusion through the capillaries to the material surface, as well as achievement by the cavitation bubble of a maximum size equal to the capillary diameter. Moreover, the power costs for an ultrasonic impact generation at such a sound pressure level are increased significantly, and their achievement is not technically easy to implement.

Therefore, it should be considered that the ultrasonic exposure with a sound pressure level in the range from 150 to 170 dB is the most efficient (optimal) for dehydration of the capillary-porous bodies.

Further, the dispersion efficiency (the moisture evacuation rate from the entire material surface) was calculated using the cubic sample. Since in the real ultrasonic drying process there are several samples of a given size, then the energy efficiency of the process is specified by the ratio of total productivity to the sample unit volume, i.e. the amount of moisture evacuated from a sample unit volume per unit time.

The dependences of total productivity on the sample size, referred to its volume, on the sound pressure level and on the sample size are given in Figs 14 and 15, respectively.

An analysis of the obtained dependences of the dispersion productivity on the sample sizes at various sound pressure levels makes it possible to determine a significant increase in the process productivity when drying the material samples that have the well-defined dimensions, equal to the wavelength of ultrasonic vibrations in the air. With the sizes are smaller or larger than the wavelength in the air, the rate of the process is decreased.

Thus, the ultrasonic dispersion mechanism based on the cavitation phenomena was substantiated.

## Confirmation of the moisture evacuation mechanism in the droplet form

To confirm the applicability of the moisture evacuation mechanism in the droplet form under the influence of ultrasonic vibrations, the drops deposited on a glass slide with an immersion

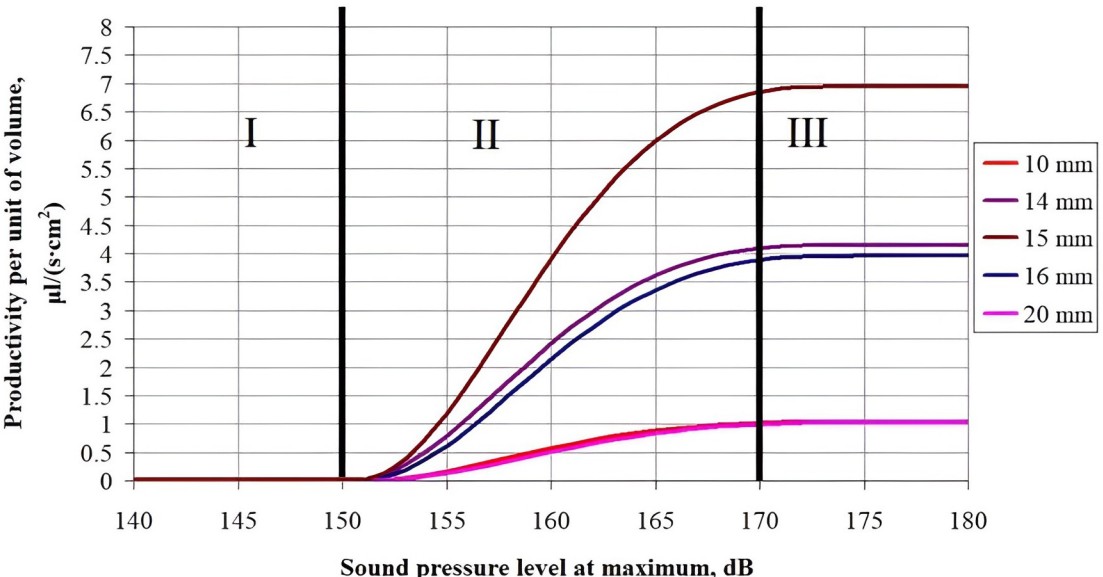

**Fig 14. Dependence of the total moisture dispersion productivity from a volume of 1 cm³ on the sound pressure level for various sample sizes.**

medium were studied. The glass was located in the immediate vicinity of the material being dried and in the outlet branch pipe of the test stand. The photographs of drops collected under ultrasonic exposure with a sound pressure level of 170 dB, depending on the time after beginning of the drying process, are shown in Fig 16.

It can be seen that with the passage of time upon commencement of drying, the number of trapped droplets is decreased. At the initial drying stage, the number of drops is significantly

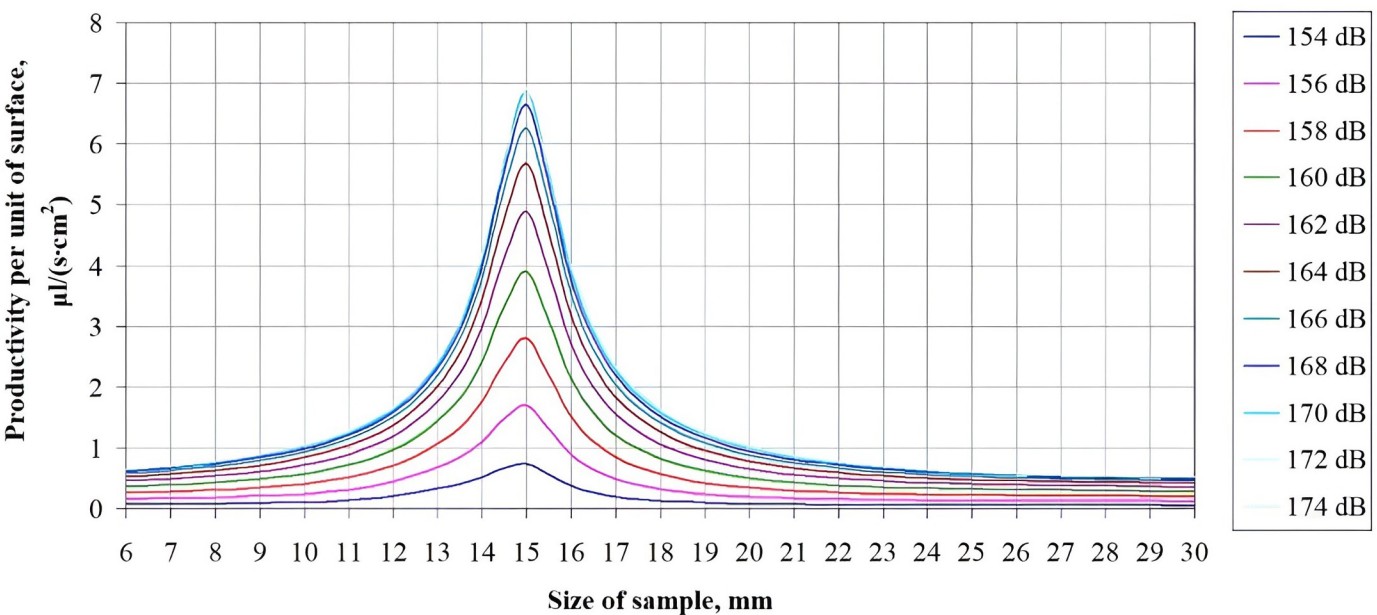

**Fig 15. Dependence of the moisture dispersion productivity from a volume of 1 cm³ on the sample size.**

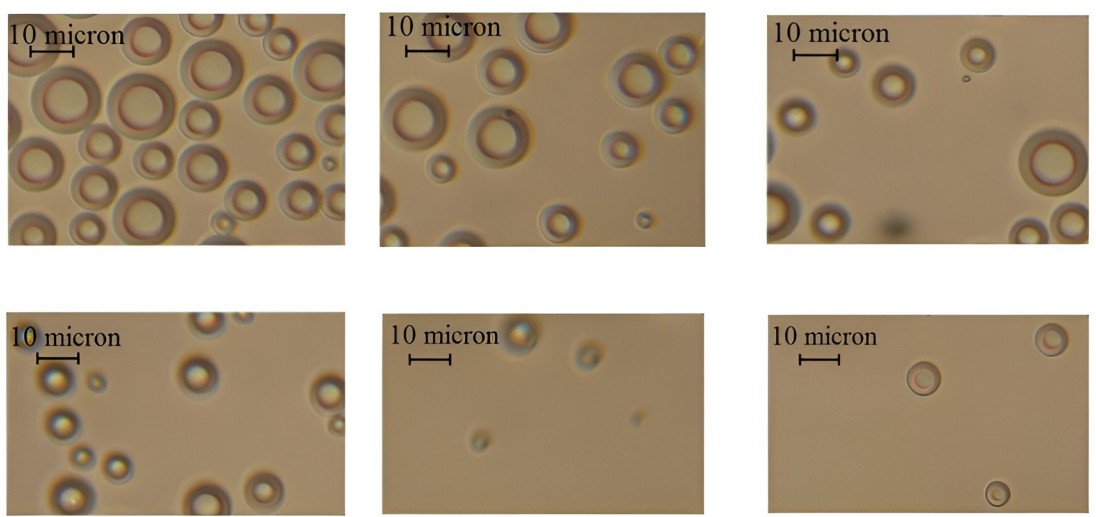

**Fig 16. Photographs of drops at various points of time that have passed since the commencement of drying process.** near the material being dried ((**a**) - 5 minutes; (**b**) - 10 minutes; (**c**) - 20 minutes) and in the outlet pipe of the test stand ((**d**) - 5 minutes; (**e**) - 10 minutes; (**f**) - 20 minutes).

greater than after 10 minutes upon the commencement of drying. In turn, the number of drops collected after 10 minutes upon the commencement of drying is greater than after 20 minutes of drying. The droplet diameter is in the range from 10 μm to 20 μm.

The number of droplets at the dryer outlet is less than in the immediate vicinity of the cork plug, and their diameter does not exceed 10 μm. This fact can be explained by the evaporation of droplets along the path of the air flow and settling on the dryer's walls.

The photographs of droplets trapped at various sound pressure levels are shown in Fig 17.

During the convection drying process, no water droplets were found on the glass slides. During the ultrasonic drying process with a sound pressure level of less than 150 dB, the liquid droplets were found in a minimum quantity. However, beginning from a sound pressure level of 150 dB, the number of droplets is grown with an increase in the sound pressure level both on the glass slide near the cork plug and in the outlet of the drying chamber (Fig 16).

This fact indicates that the liquid droplet dispersion mechanism of the dried capillary-porous material is applied under the ultrasonic impact with a sound pressure level of more than 150 dB. At the higher sound pressure levels, the number of dispersed droplets is increased, while at the lower sound pressure levels, almost no dispersed droplets are generated.

The drying curves for cork material only with warm air and with the additional impact of ultrasonic vibrations at a sound pressure level of 150 dB and 170 dB are shown in Fig 18 (S1 File).



| 150 dB | 160 dB | 170 dB | 175 dB |

**Fig 17. Photographs of dispersed drops when exposed to the ultrasonic vibrations at various sound pressure levels.**

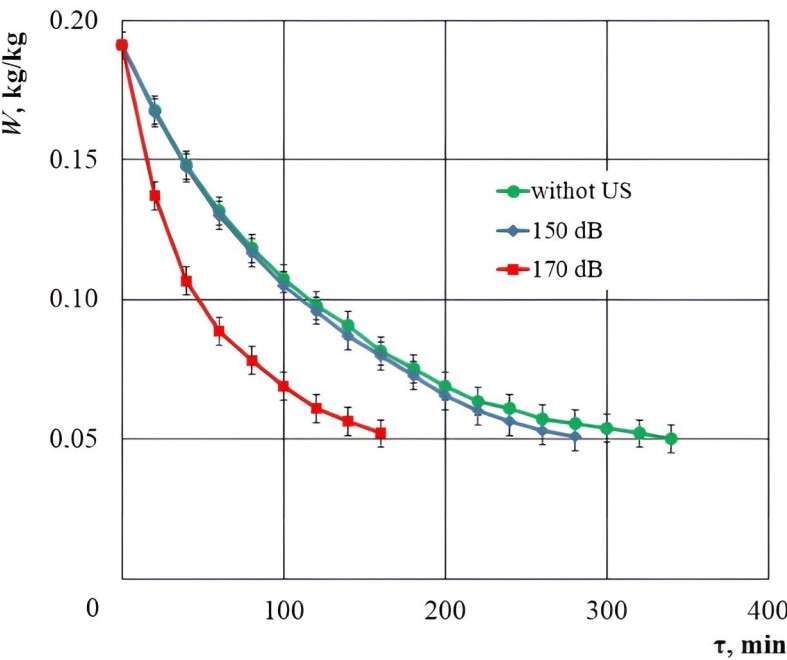

**Fig 18. Drying curves of cork plug with warm air at a temperature of $T$ = 30°C and with the additional ultrasonic exposure of 150 dB and 170 dB.** Error bars indicate standard deviation.

Theoretical regression analysis of experimental data was carried out using the exponential function $W = b \cdot \exp(-c\tau)$. The choice of function is because, according to the studies of Szadzinska [39] and Corzo [40], this is the most accurate regression model of the drying process. The function coefficients were determined using the LabPlot program (KDE Community). Coefficients $b$ and $c$, as well as the coefficient of determination $R^2$ for the approximating function are given in Table 3.

The obtained coefficient of determination $R^2$ shows a good agreement between the obtained experimental data and known theoretical models.

The duration of drying with warm air at a temperature of $T$ = 30°C, without exposure to the ultrasonic vibrations to the moisture content level of 0.05 kg/kg, was 340 min. In the case of additional exposure to the ultrasonic vibrations with a sound pressure level of 150 dB, the drying time was reduced only to 280 minutes, while at 170 dB, the drying time was equal to 160 minutes. At a sound pressure level of 150 dB or more, the changes in the structure and nature of the energy consumed for the drying process can be noted. The availability of dispersed moisture droplets and a significant reduction in the drying time prove the intense moisture evacuation from the capillary-porous body without any phase transformation.

**Table 3. Results of regression analysis of ultrasonic drying of cork with sound pressure levels of 0, 150 and 170 dB.**

| Drying process | $b$ | $c$ | $R^2$ |
|---|---|---|---|
| without US | 0.1789 | 0.00453 | 0.974 |
| with US 150 dB | 0.1831 | 0.00514 | 0.989 |
| with US 170 dB | 0.1746 | 0.00938 | 0.945 |

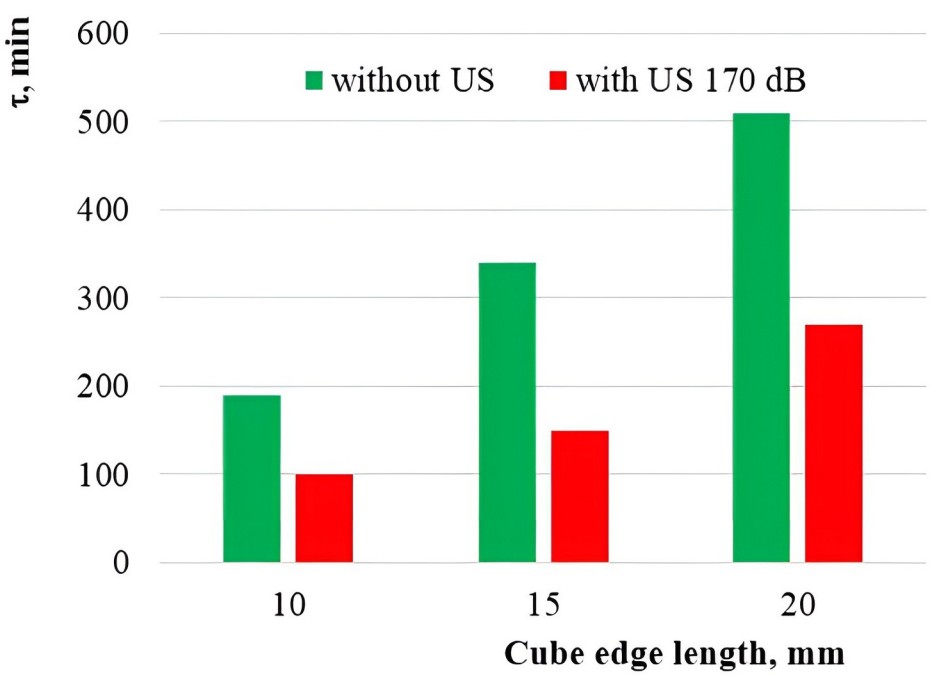

**Fig 19. Drying time histogram for various sizes of cork plugs.**

## Confirmation of the optimal material dimensions

To confirm the theoretically determined dependence of magnitude of the moisture evacuation mechanism in a droplet form on the geometric dimensions of samples of the capillary-porous material, the experiments were performed for drying the cork plugs of various sizes. The drying time histograms for the cork cubes of various sizes are shown in Fig 19 (S1 File).

With an increase in the sample dimensions, the duration of drying with warm air and combined impact of ultrasonic vibrations is also increased. However, in the case of optimal sample sizes (the edge length of 15 mm), reduction in the drying time (ultrasound drying + convection drying) compared to the convection drying only is 2.3 times, while for other sample sizes (10 mm and 20 mm) it does not exceed 1.9 times. The performed experimental studies confirm the high efficiency of ultrasonic impact on the samples of dried material with optimal size, equal to the wavelength of ultrasonic vibrations in the air.

## Experimental confirmation of the threshold values of the sound pressure level

To confirm the theoretically obtained dependence of the moisture evacuation efficiency without a phase transformation, the experiments were performed on drying the cork plug samples with the optimal dimensions (the cube edge length of 15 mm) with various sound pressure levels (140, 150, 160, 170, 175 dB). The cork material drying curves are shown in Fig 20 (S1 File).

Coefficients $b$ and $c$ and coefficient of determination $R^2$ for the approximating function $W = b \cdot \exp(-c\tau)$ are given in Table 4.

At different sound pressure levels during the drying process of cork, a high coefficient of determination $R^2$ is observed. However, at sound pressure levels of 160 dB and above, $R^2$ becomes smaller. This can serve as indirect confirmation that, with increasing sound pressure

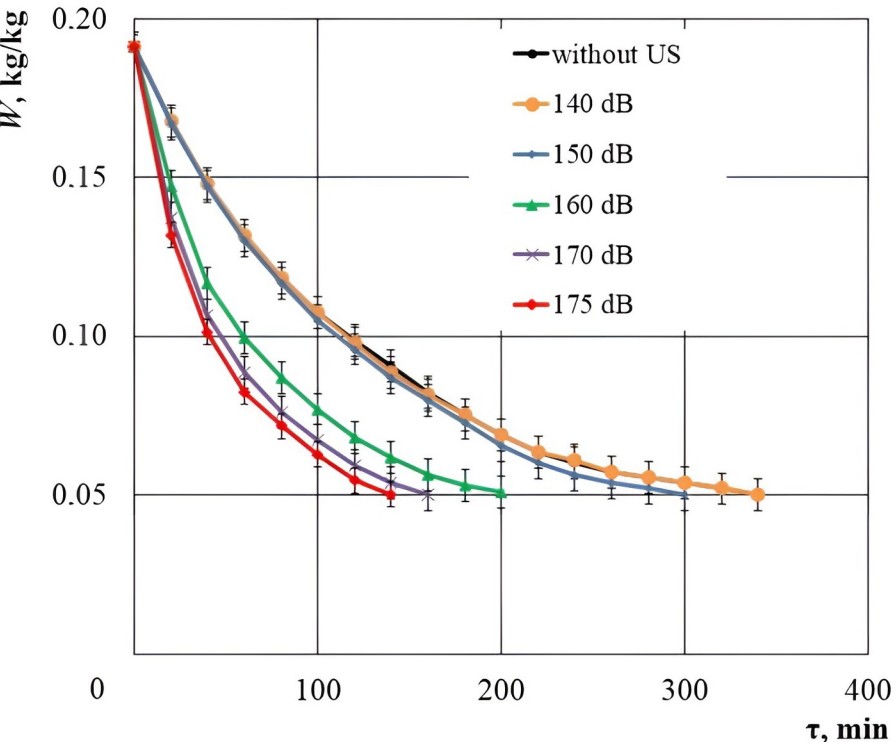

**Fig 20. Drying curves of agglomerated cork plugs at various sound pressure levels.** Error bars indicate standard deviation.

level, an increasing part of the moisture is removed due to its dispersion in droplet form (i.e., experimental data deviate more and more from the ideal model describing the process of moisture removal due to conversion to steam).

The drying duration only with a stream of warm air at a temperature of T = 30°C, to a moisture content of 0.05 kg/kg, was 340 minutes. Under the combined impact of ultrasonic vibrations with a sound pressure level of 140 dB, there was almost no reduction in the drying time. At a sound pressure level of 150 dB, there was a slight reduction in the drying time compared to the control experiment. However, already at the level of 160 dB, a significant reduction in the drying time was found. Since a significant intensification of the drying process begins at a sound pressure level of more than 150 dB, this value can be taken as a "critical sound pressure level". However, for a more detailed analysis, the dependences of the drying rate on moisture content were plotted for various process types (Fig 21, S1 File).

**Table 4. Results of regression analysis of the process of ultrasonic drying of cork with different sound pressure levels.**

| Drying process | b | c | $R^2$ |
|---|---|---|---|
| without US | 0.1790 | 0.00453 | 0.977 |
| with US 140 dB | 0.1792 | 0.00458 | 0.976 |
| with US 150 dB | 0.1820 | 0.00501 | 0.975 |
| with US 160 dB | 0.1747 | 0.00767 | 0.955 |
| with US 170 dB | 0.1757 | 0.00971 | 0.953 |
| with US 175 dB | 0.1767 | 0.01103 | 0.952 |

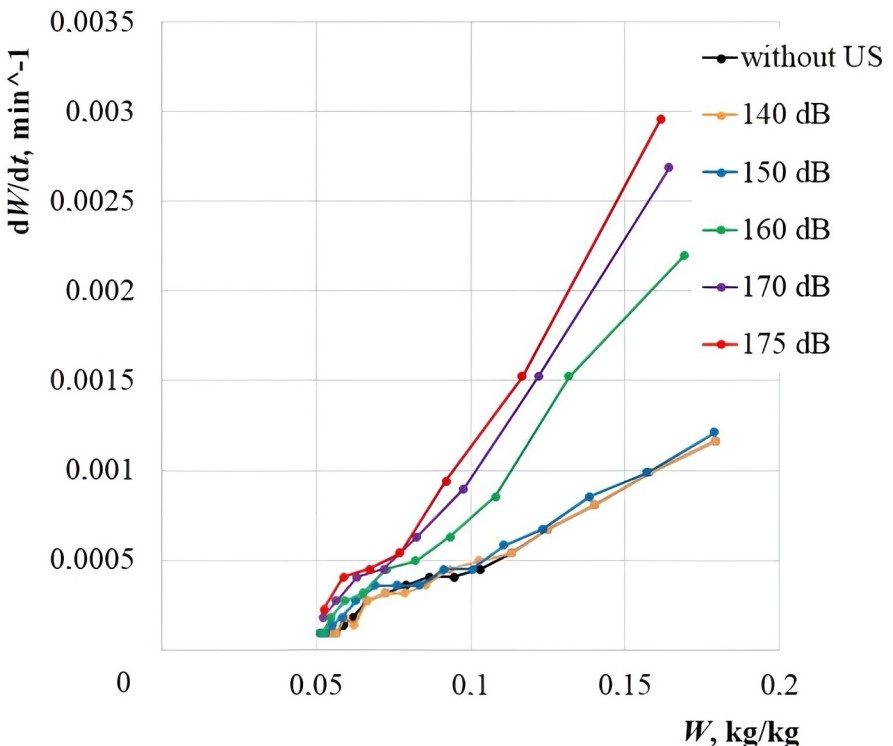

**Fig 21. Dependences of the cork plug drying rate on moisture content at various sound pressure levels.**

As expected, at the sound pressure levels up to 150 dB, there was no significant increase in the drying rate. However, with an increase in the sound pressure level of more than 150 dB, the drying speed was increased exponentially, i.e. there was an intensive material dehydration at the initial stage of drying.

For a more detailed analysis of the sound pressure level impact on the process intensification at the high and low moisture content, the averaged dependences of the drying rate on the sound pressure level were plotted at the moisture content $W > 0.096$ kg/kg and $W < 0.096$ kg/kg. The moisture content $W = 0.096$ kg/kg is the average value of the initial moisture content Wn = 0.192 kg/kg in the experiments. It is taken to separate the dried cork into the states of "high moisture content" and "low moisture content" (Fig 22, S1 File).

The dependences presented in the graphs show that at $W > 0.096$ kg/kg, the linear increase in the drying rate is disturbed and a sharp jump in the drying rate is observed in the range of sound pressure levels of 150–170 dB. Moreover, an increase in the drying rate with an increase in the sound pressure level at the low moisture content $W < 0.096$ kg/kg complies with a linear law over the entire range of sound pressure levels studied. The experimental data qualitatively comply with the theoretical data for the samples in the form of cubes with a face length equal to the wavelength of ultrasonic vibrations in the air. This fact confirms the proposed and developed theory of ultrasonic drying with the implemented cavitation dispersion. Thus, it has been established that the range of sound pressure levels of 165–170 dB is the most efficient one for the drying process intensification of capillary-porous materials with the acceptable power costs for generating ultrasonic vibrations.

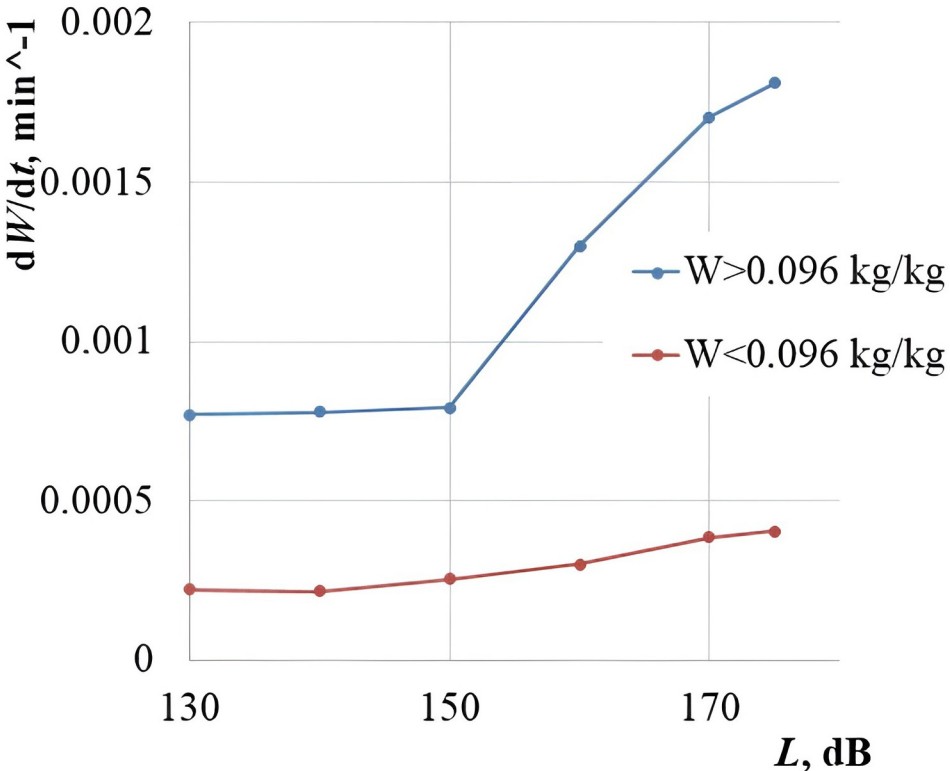

**Fig 22. Dependence of the average drying rate of the cork plug on the sound pressure level.**

### Validation of the proposed moisture evacuation mechanism for the food products

To confirm the practical applicability of the proposed moisture evacuation mechanism to the capillary-porous food products, the experiments were performed for drying of red beets.

The drying curves of red beets cut into the cubes of optimal sizes are shown in Fig 23 (S1 File).

Coefficients $b$ and $c$ and coefficient of determination $R^2$ for the approximating function $W = b \cdot \exp(-c\tau)$ are given in Table 5.

Under different conditions of the beet drying process, the highest coefficient of determination $R^2 > 0.999$ is observed. At the same time, for ultrasonic drying the coefficient of determination turns out to be slightly lower than for convective drying. As already mentioned, this is because part of the moisture is removed due to its dispersion in droplet form, which leads to a deviation of the experimental data from the drying curve described by the exponential expression.

It follows from the presented dependencies that during the convection drying with parallel and perpendicular arrangement of pallets with the material to be dried relative to the air flow direction, the drying time is equal. This fact confirms invariance of the hydrodynamic drying parameters in the case of various arrangement of pallets with the material to be dried.

In turn, if there is ultrasonic exposure, the location of pallets relative to the direction of ultrasonic vibrations affects the drying process duration. Thus, in the case of a parallel arrangement of pallets, the drying time is reduced by 1.9 times, and in the case of a perpendicular

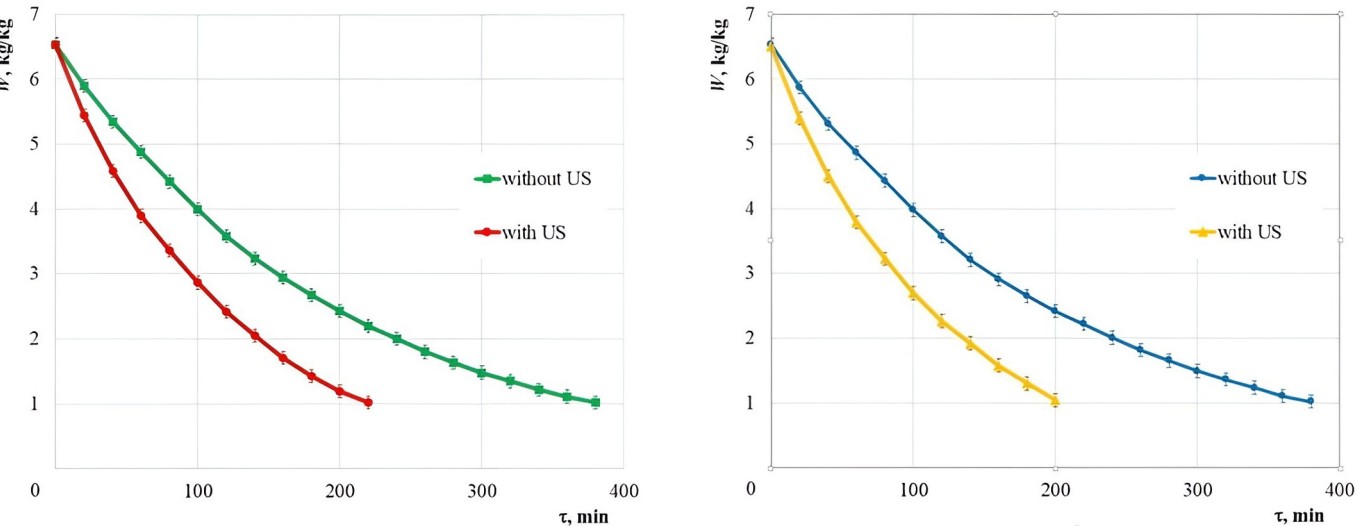

**Fig 23. Drying curves of red beets at 50°C.** Error bars indicate standard deviation. **(a)** perpendicular direction of the air flow and ultrasonic vibrations relative to the pallets; **(b)** parallel direction of air flow and ultrasonic vibrations relative to the pallets.

arrangement of pallets, it is reduced by 1.7 times relative to the drying duration only with a stream of warm air. This is probably due to the absorption of a part of ultrasonic energy in the layers located next to the disc emitter.

Other factors being equal, the location of pallets parallel to the direction of ultrasonic vibrations leads to a more noticeable reduction in the drying time than when the pallets are located perpendicular to the propagation of ultrasonic vibrations.

## Determination of the energy efficiency of ultrasonic drying process

To compare the power costs for ultrasonic drying compared to the convection one, the measurements of electric power consumption were performed with a parallel arrangement of pallets at an air flow temperature of 50°C.

The amount of electric energy consumed for ultrasonic drying was 6.5 kW·h, and the amount of electric energy consumed for convection drying was 11.4 kW·h. Hence, the energy efficiency was 43%, i.e. the electric energy consumed during the ultrasonic drying is 1.7 times less due to the moisture evacuation in a droplet form and a decrease in the process duration.

## Conclusions

As a result of the theoretical and experimental studies of the ultrasonic drying process, the following results have been obtained:

**Table 5. Results of regression analysis of the beet drying process with parallel and perpendicular ultrasonic influence.**

| Drying process | Direction of influence relative to layers of material being dried | $b$ | $c$ | $R^2$ |
|---|---|---|---|---|
| without US | perpendicular | 6.5252 | 0.00495 | 0.9999 |
| with US 165 dB | perpendicular | 6.4776 | 0.00834 | 0.9995 |
| without US | parallel | 6.5252 | 0.00496 | 0.9999 |
| with US 165 dB | parallel | 6.4752 | 0.00886 | 0.9996 |

1. The ultrasonic drying mechanism without a phase transformation of water into steam due to the liquid cavitation dispersion has been proposed and justified. Dispersion is performed by the collapse of cylindrical cavitation bubbles developed in the material capillaries.

2. The model of the moisture evacuation mechanism due to the collapse of cylindrical cavitation bubbles has been developed and an analysis of this mechanism has been performed. It has been found that a significant reduction in the drying time (an increase in its velocity) is provided by exposure to the ultrasonic vibrations at a sound pressure level of more than 150 dB ("critical level") at which the mechanism of ultrasonic water dispersion from the capillary-porous material is commenced.

3. The need to consider the full life cycle of a bubble consisting of three stages (slow growth, rapid expansion with deformation and subsequent collapse) when calculating the moisture dispersion efficiency under the impact of ultrasonic vibrations has been established.

4. A significant contribution of each of the three bubble growth stages has been established (slow growth that allows the nucleus to reach a size twice that of the capillary; expansion with deformation and collapse).

5. It has been theoretically proven that there is a limiting sound pressure level at which the growth of dispersion efficiency is ended (due to the fact that in some of the capillaries, the cavitation bubbles reach a maximum size equal to the capillary size, and the collapse process does not occur. In the case of further raise of the sound pressure level, the proportion of such capillaries is increased and hinders the growth of dispersion efficiency).

6. In theoretical terms, the optimal range of sound pressure levels of 150–170 dB has been identified, in which the dependence of dispersion efficiency on the sound pressure level is close to the linear one.

7. Based on scaling of the proposed cavitation drop generation model per unit surface area, the optimal size of the dehydrated sample is determined being relevant to the ultrasonic wavelength in the gas phase, at which the dispersion efficiency is the highest.

8. The proposed mechanism of ultrasonic liquid dispersion during the drying process has been experimentally confirmed by direct measurements of the generated liquid particles, both near the material being dried and at the outlet of the drying volume.

9. It has been experimentally found that in order to significantly reduce the drying time (up to 50% or more), it is necessary to ensure the impact of ultrasonic vibrations at a sound pressure level in the range of 165–170 dB on the materials being dried, placed in the form of particles or layers, having the dimensions or thicknesses relevant to the wavelength of ultrasonic vibrations in the air.

10. To confirm the results of theoretical studies and determine the requirements for the ultrasonic drying process in real conditions, a test stand was developed based on a disc-shaped emitter.

11. The conducted studies have made it possible to determine that at a high moisture content, an exponential increase in the average drying rate is observed in the range of sound pressure levels of 150–170 dB. Moreover, at a low moisture level, the sharp changes in the average drying rate with an increase in the sound pressure level have not been found.

12. By the example of red beet drying, it has been established that the ultrasonic exposure with a sound pressure level of 165 dB provides a reduction in the drying time by 1.9 times

compared to the convection one, hereby confirming the prospect of industrial application of ultrasonic drying for the food products.

## Supporting information

**S1 File. Minimal data set for plotting.**
(XLSX)

## Author Contributions

**Conceptualization:** Vladimir Khmelev, Andrey Shalunov.

**Data curation:** Andrey Shalunov.

**Formal analysis:** Roman Golykh.

**Funding acquisition:** Viktor Nesterov.

**Investigation:** Sergey Terentiev, Viktor Nesterov.

**Methodology:** Vladimir Khmelev, Andrey Shalunov, Roman Golykh.

**Project administration:** Vladimir Khmelev, Andrey Shalunov.

**Resources:** Viktor Nesterov.

**Software:** Roman Golykh.

**Validation:** Andrey Shalunov, Sergey Terentiev.

**Writing – original draft:** Sergey Terentiev, Roman Golykh.

**Writing – review & editing:** Vladimir Khmelev, Andrey Shalunov.

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
