## [Decision Letter · Decision Letter 0]

6 Mar 2024

PONE-D-23-23088Experimental and computer study of the mechanism and identification of conditions for energy-efficient removal of moisture from materials under ultrasonic exposurePLOS ONE

Dear Dr. Shalunov,

Thank you for submitting your manuscript to PLOS ONE. After careful consideration, we feel that it has merit but does not fully meet PLOS ONE’s publication criteria as it currently stands. Therefore, we invite you to submit a revised version of the manuscript that addresses the points raised during the review process.

We look forward to receiving your revised manuscript.

Kind regards,

Venkatasubramanian Sivakumar

Academic Editor

PLOS ONE

Journal Requirements:

“The study was supported by a grant from Russian Science Foundation No. 21-79-10359, https://rscf.ru/en/project/21-79-10359/”

5. We note that your Data Availability Statement is currently as follows: [All relevant data are within the manuscript and its Supporting Information files.]

Additional Editor Comments (if provided):

You may revise the Manuscript as per the comments pof the Reviewer and submit the revised Manuscript.

Reviewers' comments:

Reviewer's Responses to Questions

**Comments to the Author**

1. Is the manuscript technically sound, and do the data support the conclusions?

Reviewer #1: Yes

2. Has the statistical analysis been performed appropriately and rigorously? 

Reviewer #1: No

3. Have the authors made all data underlying the findings in their manuscript fully available?

Reviewer #1: Yes

4. Is the manuscript presented in an intelligible fashion and written in standard English?

Reviewer #1: Yes

5. Review Comments to the Author

Reviewer #1: The drying efficiency enhancement mechanism under ultrasonic exposure was investigated. The paper is acceptable, however, the organization should be improved.

1. The introduction part is divided into too many paragraphs, and the novelty and highlights are not well presented. A proper background knowledge of the area should be presented and what are the shortcomings or gaps in previous research? what inspiration does previous research have for this study?

2. I think the experimental results should be better analyzed by the theoretical study.

3. Ec is the amount of electric energy consumed for the convection drying, however, the electric energy consumed for the ultrasonic drying is denoted by Ec+US, this symbol can easily be misunderstood by readers.

6. PLOS authors have the option to publish the peer review history of their article (what does this mean?). If published, this will include your full peer review and any attached files.

Reviewer #1: No

---

## [Author Response · Author response to Decision Letter 0]

3 Apr 2024

Many thanks to the reviewer for his valuable comments on the article. My co-authors and I place great value on your comments. We have made detailed changes according to the comments and marked them in green in the revised manuscript.

Below are the reviewer's questions and their answers:

1. The introduction part is divided into too many paragraphs, and the novelty and highlights are not well presented. A proper background knowledge of the area should be presented and what are the shortcomings or gaps in previous research? What inspiration does previous research have for this study?

Answer. Thanks for your recommendations. We have significantly revised the Introduction: we have removed redundant information, focused attention on the problem, and indicated the works that inspired our research.

2. I think the experimental results should be better analyzed by the theoretical study.

Answer. My co-authors and I would like to express our gratitude for giving us the opportunity to finalize the article. We conducted a regression analysis of the experimental data obtained during the drying process. An exponential function was chosen as a mathematical model. The obtained model coefficients and coefficients of determination are presented in Tables 3-5.

3. Ec is the amount of electric energy consumed for the convection drying, however, the electric energy consumed for the ultrasonic drying is denoted by Ec+US, this symbol can easily be misunderstood by readers.

Answer. We are grateful that you drew attention to this annoying shortcoming. We have corrected the designation of electricity spent on ultrasonic drying.

We tried to take into account and correct all your comments. We sincerely hope that our response met your expectations.

Best wishes!

---

## [Decision Letter · Decision Letter 1]

9 Apr 2024

Experimental and computer study of the mechanism and identification of conditions for energy-efficient removal of moisture from materials under ultrasonic exposure

PONE-D-23-23088R1

Dear Dr. Shalunov,

We’re pleased to inform you that your manuscript has been judged scientifically suitable for publication and will be formally accepted for publication once it meets all outstanding technical requirements.

Kind regards,

Venkatasubramanian Sivakumar

Academic Editor

PLOS ONE

Additional Editor Comments (optional):

As Authors have addressed as per the comments of the Reviewer, the Manuscript may be accepted.

Reviewers' comments:

Reviewer's Responses to Questions

**Comments to the Author**

1. If the authors have adequately addressed your comments raised in a previous round of review and you feel that this manuscript is now acceptable for publication, you may indicate that here to bypass the “Comments to the Author” section, enter your conflict of interest statement in the “Confidential to Editor” section, and submit your "Accept" recommendation.

Reviewer #1: All comments have been addressed

2. Is the manuscript technically sound, and do the data support the conclusions?

Reviewer #1: Yes

3. Has the statistical analysis been performed appropriately and rigorously? 

Reviewer #1: Yes

4. Have the authors made all data underlying the findings in their manuscript fully available?

Reviewer #1: Yes

5. Is the manuscript presented in an intelligible fashion and written in standard English?

Reviewer #1: Yes

6. Review Comments to the Author

Reviewer #1: (No Response)

7. PLOS authors have the option to publish the peer review history of their article (what does this mean?). If published, this will include your full peer review and any attached files.

Reviewer #1: No

---

## [Editor Report · Acceptance letter]

26 Apr 2024

PONE-D-23-23088R1 

PLOS ONE

Dear Dr. Shalunov, 

I'm pleased to inform you that your manuscript has been deemed suitable for publication in PLOS ONE. Congratulations! Your manuscript is now being handed over to our production team.

Kind regards, 

on behalf of

Dr. Venkatasubramanian Sivakumar 

Academic Editor

PLOS ONE